# ENHANCING MULTIVARIATE TIME SERIES FORECASTING WITH MUTUAL INFORMATION-DRIVEN CROSS-VARIABLE AND TEMPORAL MODELING

## ABSTRACT

Recent researches have showcased the significant effectiveness of deep learning techniques for multivariate time series forecasting (MTSF). Broadly speaking, these techniques are bifurcated into two categories: Channel-independence and Channel-mixing approaches. While Channel-independence models have generally demonstrated superior outcomes, Channel-mixing methods, especially when dealing with time series that display inter-variable correlations, theoretically promise enhanced performance by incorporating the correlation between variables. However, we contend that the unnecessary integration of information through Channel-mixing can curtail the potential enhancement in MTSF model performance. To substantiate this claim, we introduce the **C**ross-variable **D**ecorrelation **A**ware feature **M**odeling (CDAM) for Channel-mixing approaches. This approach is geared toward reducing superfluous information by minimizing the mutual information between the latent representation of a single univariate sequence and its accompanying multivariate sequence input. Concurrently, it optimizes the joint mutual information shared between the latent representation, its univariate input, and the associated univariate forecast series. Notably, prevailing techniques directly project future series using a single-step forecaster, sidelining the temporal correlation that might exist across varying timesteps in the target series. Addressing this gap, we introduce the **T**emporal correlation **A**ware **M**odeling (TAM). This strategy maximizes the mutual information between adjacent subsequences of both the forecasted and target series. By synergizing CDAM and TAM, we sculpt a pioneering framework for MTSF, named as InfoTime. Comprehensive experimental analysis have demonstrated the capability of InfoTime to consistently outpace existing models, encompassing even those considered state-of-the-art.

## 1 INTRODUCTION

Multivariate time series forecasting (MTSF) plays a pivotal role in diverse applications ranging from traffic flow estimation (Bai et al., 2020), weather prediction (Chen et al., 2021), energy consumption (Zhou et al., 2021) and healthcare (Bahadori & Lipton, 2019). Deep learning has ushered in a new era for MTSF, with methodologies rooted in RNN-based (Franceschi et al., 2019; Liu et al., 2018; Salinas et al., 2020; Rangapuram et al., 2018) and CNN-based models (Lea et al., 2017; Lai et al., 2018), that surpass the performance metrics set by traditional techniques (Box et al., 2015). A notable breakthrough has been the advent of Transformer-based models (Li et al., 2019; Zhou et al., 2021; Chen et al., 2021; Zhou et al., 2022). Equipped with attention mechanisms, these models adeptly seize long-range temporal dependencies, establishing a new benchmark for forecasting efficacy. While their primary intent is to harness multivariate correlations, recent research indicates a potential shortcoming: these models might not sufficiently discern cross-variable dependencies (Murphy & Chen, 2022; Nie et al., 2022; Zeng et al., 2022). This has spurred initiatives to tease out single variable information for more nuanced forecasting.

When it comes to modeling variable dependencies, MTSF models can be broadly classified into two categories: Channel-mixing models and Channel-independence models, as highlighted in Figure 1 (a) (Nie et al., 2022). Specifically, Channel-mixing models ingest all features from the

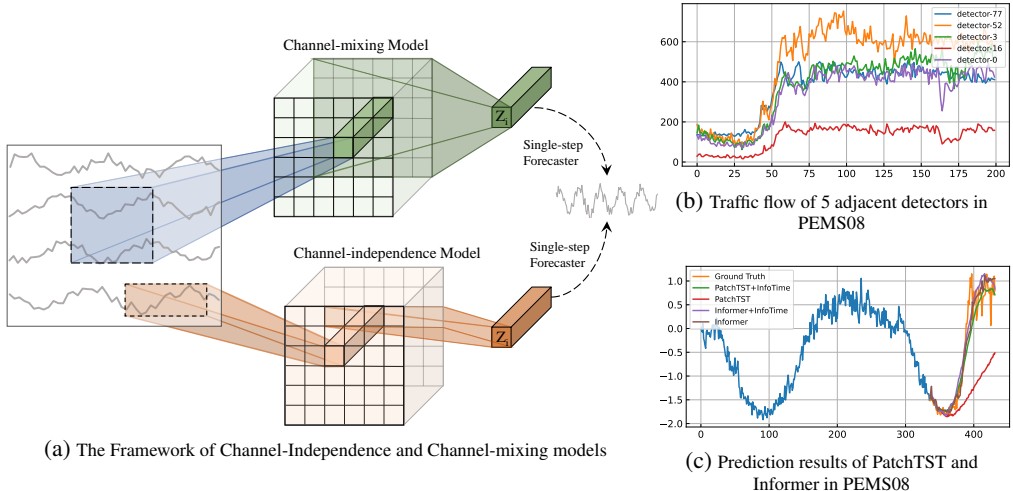

(b) Traffic flow of 5 adjacent detectors in PEMS08

(a) The Framework of Channel-Independence and Channel-mixing models

(c) Prediction results of PatchTST and Informer in PEMS08

Figure 1: (a) The framework of Channel-independence models and Channel-mixing models. Given historical series $X = \{X^i\}$ where $i$ denotes the channel index, the Channel-mixing model tends to maximize the mutual information between $X$ and the latent representation $Z^i$, and the mutual information between $Z^i$ and the i-th future series $Y^i$. The Channel-independence models maximize the mutual information between the i-th historical series $X^i$ and $Z^i$ while ignoring the mutual information between $Z^i$ and other channels; (b) Traffic flow of 5 adjacent detectors in the PEMS08 dataset; and (c) Prediction results of Channel-independence model (PatchTST), Channel-mixing model(Informer), and that with our framework, respectively.

time series, projecting them into an embedding space to blend information. Conversely, Channel-independence models restrict their input token to information sourced from just one channel. Recent studies (Murphy & Chen, 2022; Nie et al., 2022; Zeng et al., 2022) indicates that Channel-independence models significantly outpace Channel-mixing models on certain datasets. Yet, this advantage comes with a trade-off: the omission of crucial cross-variable information. Such an omission can be detrimental, especially when the variables inherently correlate. Illustratively, Figure 1 (b) showcases traffic flow variations from six proximate detectors in the PEMS08 dataset (Chen et al., 2001) . A discernible trend emerges across these detectors, suggesting that exploiting their inter-related patterns could bolster predictive accuracy for future traffic flows. In a comparative experiment, we trained both a Channel-independence model (PatchTST) and a Channel-mixing model (Informer) using the PEMS08 dataset. The outcome, as visualized in Figure 1 (c), unequivocally shows Informer's superior performance over PatchTST, underscoring the importance of cross-variable insights. Motivated by these findings, we introduce the **C**ross-Variable **D**ecorrelation **A**ware Feature **M**odeling (CDAM) for Channel-mixing methodologies. CDAM aims to hone in on cross-variable information and prune redundant data. It achieves this by minimizing mutual information between the latent depiction of an individual univariate time series and related multivariate inputs, while concurrently amplifying the shared mutual information between the latent model, its univariate input, and the subsequent univariate forecast.

Apart from modeling channel dependence, another significant challenge in MTSF is the accumulation of errors along time, as shown in Figure 1 (a). To mitigate this, a number of studies (Nie et al., 2022; Zeng et al., 2022; Zhou et al., 2021; Zhang & Yan) have adopted a direct forecasting strategy using a single-step forecaster that generates multi-step predictions in a single step, typically configured as a fully-connected network. Although often superior to auto-regressive forecasters, this method tends to neglect the temporal correlations across varied timesteps in the target series, curtailing its potential to capture series inter-dependencies effectively. Drawing inspiration from the notable temporal relationships observed in adjacent sub-sequences post-downsampling (Liu et al., 2022a), we propose **T**emporal Correlation **A**ware **M**odeling (TAM), which iteratively down-samples and optimizes mutual information between consecutive sub-sequences of both the forecasted and target series.

In essence, this paper delves into two pivotal challenges in multivariate time series forecasting: **cross-variable relationships** and **temporal relationships**. Drawing inspiration from these challenges, we develop a novel framework, denoted as InfoTime. This framework seamlessly integrates CDAM and TAM. Our paper's key contributions encompass:

- We introduce **C**ross-Variable **D**ecorrelation **A**ware Feature **M**odeling (CDAM) designed specifically for Channel-mixing methods. It adeptly distills cross-variable information, simultaneously filtering out superfluous information.
- Our proposed **T**emporal Correlation **A**ware **M**odeling (TAM) is tailored to effectively capture the temporal correlations across varied timesteps in the target series.
- Synthesizing CDAM and TAM, we unveil a cutting-edge framework for MTSF, denominated as InfoTime.

Through rigorous experimentation on diverse real-world datasets, it's evident that our InfoTime consistently eclipses existing Channel-mixing benchmarks, achieving superior accuracy and notably mitigating overfitting. Furthermore, InfoTime enhances the efficacy of Channel-Independent models, especially in instances with ambiguous cross-variable traits.

## 2 RELATED WORK

### 2.1 MULTIVARIATE TIME SERIES FORECASTING

Multivariate time series forecasting is the task of predicting future values of variables, given historical observations. With the development of deep learning, various neural models have been proposed and demonstrated promising performance in this task. RNN-based (Franceschi et al., 2019; Salinas et al., 2020; Rangapuram et al., 2018) and CNN-based (Lea et al., 2017; Lai et al., 2018) models are proposed for models time series data using RNN or CNN respectively, but these models have difficulty in modeling long-term dependency. In recent years, a large body of works try to apply Transformer models to forecast long-term multivariate series and have shown great potential (Li et al., 2019; Zhou et al., 2021; Chen et al., 2021; Zhou et al., 2022; Nie et al., 2022), Especially, LogTrans (Li et al., 2019) proposes the LogSparse attention in order to reduce the complexity from $O(L^2)$ to $O(L(\log L)^2)$. Informer (Zhou et al., 2021) utilizes the sparsity of attention score through KL-divergence estimation and proposes ProbSparse self-attention mechanism which achieves $O(L \log L)$ complexity. Autoformer (Chen et al., 2021) introduces a decomposition architecture with the Auto-Correlation mechanism to capture the seasonal and trend features of historical series which also achieves $O(L \log L)$ complexity and has a better performance. Afterword, FEDformer (Zhou et al., 2022) employs the mixture-of-expert to enhance the seasonal-trend decomposition and achieves $O(L)$ complexity. The above methods focus on modeling temporal dependency yet omit the correlation of different variables. Crossformer (Zhang & Yan) introduces Two-Stage Attention to effectively capture the cross-time and cross-dimension dependency. Recently, several works (Murphy & Chen, 2022; Nie et al., 2022; Zeng et al., 2022) observe that modeling cross-dimension dependency makes neural models suffer from overfitting in most benchmarks, therefore, they propose Channel-Independence methods to avoid this issue. However, the improvement is based on the sacrifice of cross-variable information. Besides, existing models primarily focus on extracting correlations of historical series while disregarding the correlations of target series.

### 2.2 MUTUAL INFORMATION AND INFORMATION BOTTLENECK

Mutual Information (MI) is an entropy-based measure that quantifies the dependence between random variables which has the form:

$$I(X;Y) = \int p(x,y) log \frac{p(x,y)}{p(x)p(y)} dxdy = E_{p(x,y)} \left[ log \frac{p(x,y)}{p(x)p(y)} \right] \tag{1}$$

Mutual Information was used in a wide range of domains and tasks, including feature selection (Kwak & Choi, 2002), causality (Butte & Kohane, 1999), and Information Bottleneck (Tishby et al., 2000). Information Bottleneck (IB) was first proposed by Tishby et al. (2000) which is an information theoretic framework for extracting the most relevant information in the relationship of

the input with respect to the output, which can be formulated as $max\ I(Y;Z) - \beta I(X;Z)$. Several works (Tishby & Zaslavsky, 2015; Shwartz-Ziv & Tishby, 2017) try to use the Information Bottleneck framework to analyze the Deep Neural Networks by quantifying Mutual Information between the network layers and deriving an information theoretic limit on DNN efficiency. Variational Information Bottleneck (VIB) was also proposed (Alemi et al., 2016) to bridge the gap between Information Bottleneck and deep learning. In recent years, many lower-bound estimations (Belghazi et al., 2018; Oord et al., 2018) and upper-bound estimations (Poole et al., 2019; Cheng et al., 2020) have been proposed to estimate MI effectively which are useful to estimate VIB. Nowadays, MI and VIB have been widely used in computer vision (Schulz et al., 2020; Luo et al., 2019), natural language processing (Mahabadi et al., 2021; West et al., 2019; Voita et al., 2019), reinforcement learning (Goyal et al., 2019; Igl et al., 2019), and representation learning (Federici et al., 2020; Hjelm et al., 2018). However, Mutual Information and Information Bottleneck are less researched in Multivariate Long-term series forecasting.

## 3 METHOD

In multivariate time series forecasting, one aims to predict the future value of time series $y_t = s_{t+T+1:t+T+P} \in R^{P \times C}$ given the history $x_t = s_{t:t+T} \in R^{T \times C}$, where $T$ and $P$ is the number of time steps in the past and future. $C \geq 1$ is the number of variables. Given time series $s$, we divide it into history set $X = \{x_1, ..., x_N\}$ and future set $Y = \{y_1, ..., y_N\}$, where $N$ is the number of samples. As shown in Figure 1 (a), deep learning methods first extract latent representation $Z^i$ from $X$ (Channel-mixing), or $X^i$ (Channel-independent), and then generate target series $Y^i$ from $Z^i$. A natural assumption is that these $C$ series are associated which helps to improve the forecasting accuracy. Therefore, to utilize the cross-variable dependencies while eliminating superfluous information, in Section 3.1, we propose the **C**ross-Variable **D**ecorrelation **A**ware Feature **M**odeling (CDAM) to extract cross-variable dependencies. In section 3.2, we introduce **T**emporal **A**ware **M**odeling (TAM) to predict the future series.

### 3.1 CROSS-VARIABLE DECORRELATION AWARE FEATURE MODELING

Recent studies (Nie et al., 2022; Zeng et al., 2022; Zhou et al., 2021; Zhang & Yan) have demonstrated that Channel-independence is more effective in achieving high-level performance than Channel-mixing. However, multivariate time series contain correlations among variables. Channel-mixing aims to take advantage of these cross-variable dependencies to predict future series. In fact, it fails to improve the performance of MTSF. This may be because Channel-mixing introduces that superfluous information. To verify this, we introduce CDAM to extract cross-variable information while eliminating superfluous information. Specifically, inspired information bottlenecks, CDAM maximizes the joint mutual information among the latent representation $Z^i$, its univariate input $X^i$ and the corresponding univariate target series $Y^i$ while minimizing the mutual information between latent representation $Z^i$ of one single univariate time series and other multivariate series input $X^o$. Thus, we have the objective:

$$\max\ I(Y^i, X^i; Z^i)\ s.t.\ I(X^o; Z^i) \leq I_c, \tag{2}$$

where $I_c$ is the information constraint, $X$ is the set of multivariate historical series, $X^i$ is the historical series of $i$-th variable , $X^o$ is the other multivariate series, $Z^i \in R^d$ is the representation of $X^i$ via mixing $X^o$ and used to predict the $i$-th future series $Y^i$.

With the introduction of a Lagrange multiplier $\beta$, we can maximize the objective function for $i$-th channel:

$$\begin{aligned} \mathcal{R}_{IB}^i &= I(Y^i, X^i; Z^i) - \beta I(X^o; Z^i) \\ &= I(Y^i; Z^i | X^i) + I(X^i; Z^i) - \beta I(X^o; Z^i), \end{aligned} \tag{3}$$

where $\beta \geq 0$ controls the tradeoff between $I(Y^i; Z^i | X^i)$, $I(X^i; Z^i)$ and $I(X^o; Z^i)$, the larger $\beta$ corresponds to lower mutual Information between $X^o$ and $Z^i$, and also means that $Z^i$ needs to retain the important information in $X^o$ and eliminate the irrelevant information to ensure $Y^i$ can be accurately predicted. However, the Mutual Information $I(X^i, Y; Z^i)$ and $I(X^o; Z^i)$ are intractable, we now provide the variational lower bound and upper bound for $I(X^i, Y^i; Z^i)$ and $I(X^o; Z^i)$, respectively.

**Lower bound for** $I(X^i, Y^i; Z^i)$. The joint mutual information between latent representation $Z^i$, $i$-th historical series $X^i$, and $i$-th target series $Y^i$ is defined as (More details are shown in Appendix A.3.1):

$$
\begin{aligned}
I(X^i, Y^i; Z^i) &= I(Z^i; X^i) + I(Z^i; Y^i | X^i) \\
&= \mathbb{E}_{p(z^i, y^i, x^i)} \left[ \log p(y^i | x^i, z^i) \right] + \mathbb{E}_{p(z^i, x^i)} \left[ \log p(x^i | z^i) \right] + H(Y^i, X^i),
\end{aligned}
\tag{4}
$$

where the joint entropy $H(Y^i, X^i) = -\int p(y^i, x^i) dx^i dy^i$ is only related to the dataset and cannot be optimized, so can be ignored. Therefore, MI can be simplified as:

$$
I(X^i, Y^i; Z^i) = \mathbb{E}_{p(z^i, y^i, x^i)} \left[ \log p(y^i | x^i, z^i) \right] + \mathbb{E}_{p(z^i, x^i)} \left[ \log p(x^i | z^i) \right] + \text{constant}.
\tag{5}
$$

Since $p(y^i | x^i, z^i)$ and $p(x^i | z^i)$ are intractable, we introduce $p_\theta(y^i | z^i, x^i)$ and $p_\theta(x^i | z^i)$ to be the variational approximation to $p(y^i | x^i, z^i)$ and $p(x^i | z^i)$, respectively. Thus the variational lower bound is as follows (More details are shown in Appendix A.3.2):

$$
\begin{aligned}
I(X^i, Y^i; Z^i) - \text{constant} &\geq \mathbb{E}_{p(z^i, y^i, x^i)} \left[ \log p_\theta(y^i | x^i, z^i) \right] + \mathbb{E}_{p(z^i, x^i)} \left[ \log p_\theta(x^i | z^i) \right] \\
&= I_v(X^i, Y^i; Z^i).
\end{aligned}
\tag{6}
$$

Hence, we can achieve the maximization of $I(X^i, Y^i; Z^i)$ by maximizing $I_v(X^i, Y^i; Z^i)$. We assume the variational distribution $p_\theta(y^i | z^i, x^i)$ and $p_\theta(x^i | z^i)$ as the Gaussion distribution. Thus, the first term of $I_v(X^i, Y^i; Z^i)$ is the negative log-likelihood of the prediction of $Y^i$ given $Z^i$ and $X^i$, and the second term aims to the reconstruction of $X^i$ given $Z^i$.

**Upper bound for** $I(X^o; Z^i)$. Next, to minimize the MI between the latent representation $Z^i$ and historical series $X^o$, we adopt the sampled vCLUB (Cheng et al., 2020), which is defined as:

$$
I_{vCLUB-S}(X^o; Z^i) = \frac{1}{N} \sum_{n=1}^{N} \left[ \log q_\theta(z_n^i | x_n^o) - \log q_\theta(z_n^i | x_{k'_n}^o) \right],
\tag{7}
$$

where $(z_n^i, x_{k'_n}^o)$ is a negative pair and $k'_n$ is uniformly selected from indices $1, 2, ... N$. Thus we can minimize $I(X^o; Z^i)$ by minimizing $I_{vCLUB-S}(X^o; Z^i)$. It enables the model to extract useful cross-variable information while eliminating irrelevant information.

Finally, We can convert the intractable objective function $\mathcal{R}_{IB}^i$ of all channels in Eq. 3 as:

$$
\mathcal{L}_{IB} = \frac{1}{C} \sum_{i=1}^{C} \left[ -I_v(X^i, Y^i; Z^i) + \beta I_{vCLUB-S}(X^o; Z^i) \right] \geq -\frac{1}{C} \sum_{i=1}^{C} \mathcal{R}_{IB}^i.
\tag{8}
$$

## 3.2 TEMPORAL CORRELATION AWARE MODELING

To alleviate the error accumulation effects, previous works (Nie et al., 2022; Zeng et al., 2022; Zhou et al., 2021; Zhang & Yan) use a single-step forecaster which is usually a fully-connected network to predict the future series. Different from auto-regressive forecaster, single-step forecaster assumes the predicted future time steps are independent of each other given the historical time series. Then, the training objective of the single-step forecaster can be expressed as follows :

$$
p(y^i | z^i, x^i) = \prod_{j=1}^{P} p(y_j^i | z^i, x^i)
\tag{9}
$$

Although the single-step forecaster out-performs the auto-regressive forecaster, it

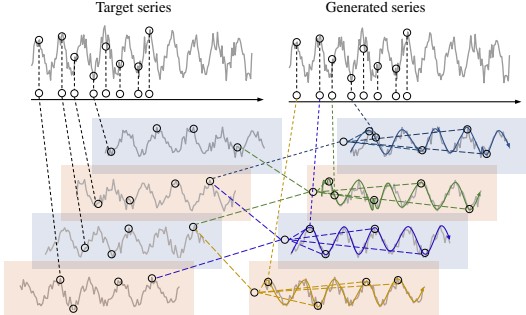

Figure 2: Architecture of TAM with 4x downsampling. We downsample the target series and forecasted series utilizing single-forecaster into four subsequences, respectively. And then we maximize the mutual information between the adjacent subsequences of forecasted series and target series.

fails to model the temporal correlations of
different timesteps in the target series. In contrast to NLP, time series data is a low-density information source (Lin et al., 2023), and one unique property of time series is that the temporal relations (e.g., the trend and the seasonal) between downsampling adjacent sub-sequences are largely preserved (Liu et al., 2022a). Based on the above observations, we propose TAM which improves the correlation of predicted future time steps by iteratively down-samples and optimizes mutual information between consecutive sub-sequences of both the forecasted and target series.

After extracting cross-variable feature $Z^i$, We first generate $\hat{Y}^i$ using a single-step forecaster by utilizing the historical data of the $i$-th channel $X^i$ and cross-variable $Z^i$, the forecasted series $\hat{Y}$ and target series $Y$ are then downsampled $N$ times. For the $n$-th ($n \leq N$) downsampling, we generate $m$ sub-sequences $\hat{Y} = \{\hat{Y}_1, ..., \hat{Y}_m\}, Y = \{Y_1, ..., Y_m\}$, where $m = 2^n$ and $\hat{Y}_j \in R^{\frac{P}{2^n} \times C}$. Then we maximize the mutual information between $\hat{Y}_j^i$ and $Y_{j-1}^i, Y_{j+1}^i$, given $X^i$, where $1 < k < m$. Therefore, the loss function of $n$-th downsampling can be calculated as:

$$\mathcal{L}_n = -\frac{1}{mC} \sum_{i=1}^{C} \left[ I(Y_2^i; \hat{Y}_1^i | X^i) + I(Y_{m-1}^i; \hat{Y}_m^i | X^i) + \sum_{j=2}^{m-1} I(Y_{j-1}^i; \hat{Y}_j^i | X^i) + I(Y_{j+1}^i; \hat{Y}_j^i | X^i) \right]$$
(10)

And the variational lower bound of $I(Y_{j-1}^i; \hat{Y}_j^i | X^i)$ is as follows (More details are shown in Appendix A.3.2):

$$I(Y_{j-1}^i; \hat{Y}_j^i | X^i) \geq \mathbb{E}_{p(y_{j-1}^i, \hat{y}_j^i, x^i)} \left[ p_\theta(y_{j-1}^i | \hat{y}_j^i, x^i) \right]$$
(11)

Furthermore, considering the efficiency, we assume that the time steps of a sub-sequence are independent given the adjacent sub-sequence. Therefore, $I(Y_{j-1}^i; \hat{Y}_j^i | X^i)$ can be simplified as $I(Y_{j-1}^i; \hat{Y}_j^i | X^i) = \sum_{k=1}^{\frac{P}{2^n}} \left[ I(Y_{j-1,k}^i; \hat{Y}_j^i | X^i) \right]$ and we can generate the entire sub-sequence in a single step without auto-regression.

For the $n$-th downsampling, TAM will generate $2 \times (2^n - 1)$ sub-sequences $\hat{Y}' = \{\hat{Y}_1^r, \hat{Y}_2^l, \hat{Y}_2^r, ..., \hat{Y}_m^l\}$, these sub-sequences that are not at ends are predicted by its left and right adjacent sub-sequences respectively. We splice these $2 \times (2^n - 1)$ sub-sequences into a new series $\hat{Y}_n = \{\hat{Y}_1^r, \frac{\hat{Y}_2^l + \hat{Y}_2^r}{2}, ..., \hat{Y}_m^l\}$. After $N$ times downsampling, we have generated $N + 1$ series. And we use these $N + 1$ series as the final forecasting results, thus we have the following loss function:

$$\mathcal{L}_p = ||Y - (\lambda \sum_{n=1}^{N} \frac{\hat{Y}_n}{N} + (1 - \lambda)\hat{Y})||_2^2$$
(12)

In contrast to single-step forecasters that generate multi-step predictions without considering the correlation between the predicted series, our proposed method, referred to as TAM, explicitly models the correlation of predicted future time steps. It achieves this by iteratively down-sampling and optimizing the mutual information between consecutive sub-sequences of both the forecasted and target series. This approach allows the model to establish more accurate representations of future sequences, thereby enhancing the overall predictive performance. By incorporating the correlation between predicted time steps, TAM considers the temporal dependencies within the forecasted series and captures the underlying patterns in the data. This iterative down-sampling and mutual information optimization procedure ensures that the model effectively leverages the available information to generate more accurate and coherent predictions.

Integrating CDAM and TAM, the total loss of InfoTime can be written as :

$$\mathcal{L}_{total} = \mathcal{L}_{IB} + \sum_{n=1}^{N} \mathcal{L}_n + \mathcal{L}_p$$
(13)

## 4 EXPERIMENTS

In this section, we extensively evaluate the proposed InfoTime on nine real-world benchmarks using various Channel-mixing and Channel-Independence models, including state-of-the-art models.

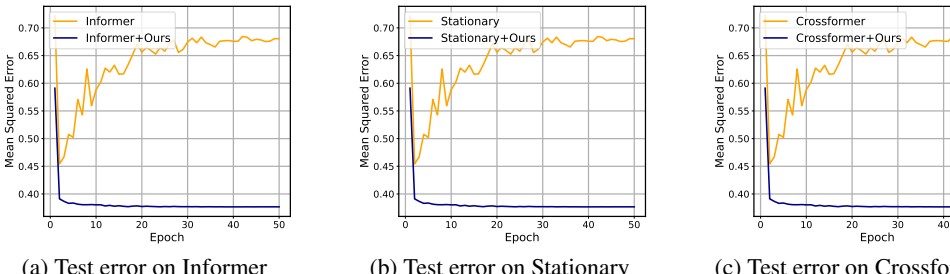

(a) Test error on Informer      (b) Test error on Stationary      (c) Test error on Crossformer

Figure 3: **Test error for each training epoch**. We train the baselines and integrated with our InfoTime for 50 epochs on ETTh1 when the history and prediction length are both 96.

**Baselines.** Since our method can be easily applied to deep-learning-based forecasting models, we evaluate InfoTime adopted by several popular baselines including the state-of-the-art method. For Channel-mixing models, we select Informer (Zhou et al., 2021), Non-stationary Transformer (Liu et al., 2022b), denoting as Stationary, and Crossformer (Zhang & Yan). For Channel-Independence models, we use PatchTST (Nie et al., 2022) and also propose RMLP which consists of two linear layers with relu activation, and also uses the reversible instance normalization (Kim et al., 2021). (see Appendix A.1 for more details)

**Datasets.** We evaluate the performance of InfoTime in nine widely-used real-world datasets. Here is a detailed description of these datasets. (1) The ETT (Zhou et al., 2021) (Electricity Transformer Temperature) dataset contains two years of data from two separate countries in China with intervals of 1-hour level (ETTh) and 15-minute level (ETTm) collected from electricity transformers. Each time step contains six power load features and oil temperature. (2) The Electricity [1] dataset describes 321 clients' hourly electricity consumption from 2012 to 2014. (3) The Traffic [2] dataset contains the road occupancy rates from various sensors on San Francisco Bay area freeways, which is provided by California Department of Transportation. (4) the Weather [3] dataset contains 21 meteorological indicators collected at around 1,600 landmarks in the United States. (5)The PEMS (Chen et al., 2001) (PEMS03, PEMS04, and PEMS08) measures the highway traffic of California in real-time every 30 seconds. We follow the standard protocol that divides each dataset into the training, validation, and testing subsets according to the chronological order. The split ratio is 6:2:2 for the ETT dataset and 7:1:2 for others.

### 4.1   MAIN RESULTS

Table 1 compares the forecasting accuracy of the Channel-mixing baselines and InfoTime. The results show that InfoTime consistently outperforms all three baselines, Informer, Stationary, and Crossformer, by a large margin. Moreover, the effectiveness of InfoTime is more evident for the long sequence prediction which may be long sequence prediction is more difficult and more likely to lead the model depending on superfluous cross-variable information. InfoTime shows a stable performance in contrast to the baselines, which show a high increase in error as prolonging the prediction length. For example, when the prediction length increases from 96 to 720 on the ETTm2 dataset, the forecasting error of Informer significantly increases from 0.365 to 3.379. In contrast, InfoTime shows a much slight increase in error. A similar tendency appears with the other prediction lengths, datasets, and baseline models as well. These results demonstrate that InfoTime makes the baseline models more robust to prediction target series. Additionally, to study how InfoTime can perform better than baselines, we visualize the testing error for each epoch in Figure 3. Overall, InfoTime shows lower and more stable test errors compared to the baselines. Moreover, the baselines are extremely prone to overfitting in the early stages of training, and InfoTime can effectively alleviate this problem.

---

[1] https://archive.ics.uci.edu/ml/datasets/ElectricityLoadDiagrams20112014.

[2] http://pems.dot.ca.gov.

[3] https://www.bgc-jena.mpg.de/wetter/.

Table 1: Multivariate long-term series forecasting results on Channel-mixing models with different prediction lengths $O \in \{96, 192, 336, 720\}$. We set the input length $I$ as 96 for all the models. The best result is indicated in bold font. *Avg* is averaged from all four prediction lengths and *Pro* means the relative MSE or MAE reduction.

| Models | | Informer | | | | Stationary | | | | Crossformer | | | |
|---|---|---|---|---|---|---|---|---|---|---|---|---|---|
| | | Original | | w/Ours | | Original | | w/Ours | | Original | | w/Ours | |
| Metric | | MSE | MAE | MSE | MAE | MSE | MAE | MSE | MAE | MSE | MAE | MSE | MAE |
| ETTh1 | 96 | 0.865 | 0.713 | **0.381** | **0.394** | 0.598 | 0.498 | **0.375** | **0.388** | 0.457 | 0.463 | **0.379** | **0.392** |
| | 192 | 1.008 | 0.792 | **0.435** | **0.430** | 0.602 | 0.520 | **0.425** | **0.417** | 0.635 | 0.581 | **0.433** | **0.427** |
| | 336 | 1.107 | 0.809 | **0.485** | **0.461** | 0.677 | 0.573 | **0.463** | **0.436** | 0.776 | 0.667 | **0.482** | **0.458** |
| | 720 | 1.181 | 0.865 | **0.534** | **0.524** | 0.719 | 0.597 | **0.463** | **0.459** | 0.861 | 0.725 | **0.529** | **0.517** |
| | Avg | 1.040 | 0.794 | **0.458** | **0.452** | 0.649 | 0.547 | **0.431** | **0.425** | 0.682 | 0.609 | **0.455** | **0.448** |
| | Pro | - | - | 55.9% | 43.0% | - | - | 33.5% | 22.3% | - | - | 33.2% | 26.4% |
| ETTh2 | 96 | 3.755 | 1.525 | **0.336** | **0.390** | 0.362 | 0.393 | **0.286** | **0.335** | 0.728 | 0.615 | **0.333** | **0.386** |
| | 192 | 5.602 | 1.931 | **0.468** | **0.470** | 0.481 | 0.453 | **0.371** | **0.388** | 0.898 | 0.705 | **0.455** | **0.453** |
| | 336 | 4.721 | 1.835 | **0.582** | **0.534** | 0.524 | 0.487 | **0.414** | **0.425** | 1.132 | 0.807 | **0.554** | **0.513** |
| | 720 | 3.647 | 1.625 | **0.749** | **0.620** | 0.512 | 0.494 | **0.418** | **0.437** | 4.390 | 1.795 | **0.757** | **0.619** |
| | Avg | 4.431 | 1.729 | **0.534** | **0.504** | 0.470 | 0.457 | **0.372** | **0.396** | 1.787 | 0.981 | **0.525** | **0.493** |
| | Pro | - | - | 87.9% | 70.9% | - | - | 20.9% | 13.3% | - | - | 70.6% | 49.7% |
| ETTm1 | 96 | 0.672 | 0.571 | **0.326** | **0.367** | 0.396 | 0.401 | **0.326** | **0.362** | 0.385 | 0.409 | **0.323** | **0.362** |
| | 192 | 0.795 | 0.669 | **0.371** | **0.391** | 0.471 | 0.436 | **0.366** | **0.379** | 0.459 | 0.478 | **0.366** | **0.386** |
| | 336 | 1.212 | 0.871 | **0.408** | **0.416** | 0.517 | 0.464 | **0.392** | **0.398** | 0.649 | 0.583 | **0.403** | **0.414** |
| | 720 | 1.166 | 0.823 | **0.482** | **0.464** | 0.664 | 0.527 | **0.455** | **0.434** | 0.756 | 0.669 | **0.473** | **0.460** |
| | Avg | 0.961 | 0.733 | **0.396** | **0.409** | 0.512 | 0.457 | **0.384** | **0.393** | 0.562 | 0.534 | **0.391** | **0.405** |
| | Pro | - | - | 58.7% | 44.2% | - | - | 25.0% | 14.0% | - | - | 30.4% | 24.1% |
| ETTm2 | 96 | 0.365 | 0.453 | **0.187** | **0.282** | 0.201 | 0.291 | **0.175** | **0.256** | 0.281 | 0.373 | **0.186** | **0.281** |
| | 192 | 0.533 | 0.563 | **0.277** | **0.351** | 0.275 | 0.335 | **0.238** | **0.297** | 0.549 | 0.520 | **0.269** | **0.341** |
| | 336 | 1.363 | 0.887 | **0.380** | **0.420** | 0.350 | 0.377 | **0.299** | **0.336** | 0.729 | 0.603 | **0.356** | **0.396** |
| | 720 | 3.379 | 1.338 | **0.607** | **0.549** | 0.460 | 0.435 | **0.398** | **0.393** | 1.059 | 0.741 | **0.493** | **0.482** |
| | Avg | 1.410 | 0.810 | **0.362** | **0.400** | 0.321 | 0.359 | **0.277** | **0.320** | 0.654 | 0.559 | **0.326** | **0.375** |
| | Pro | - | - | 74.3% | 50.6% | - | - | 13.7% | 10.8% | - | - | 50.1% | 32.9% |
| Weather | 96 | 0.300 | 0.384 | **0.179** | **0.249** | 0.181 | 0.230 | **0.166** | **0.213** | 0.158 | 0.236 | **0.149** | **0.218** |
| | 192 | 0.598 | 0.544 | **0.226** | **0.296** | 0.286 | 0.312 | **0.218** | **0.260** | 0.209 | 0.285 | **0.202** | **0.272** |
| | 336 | 0.578 | 0.523 | **0.276** | **0.334** | 0.319 | 0.335 | **0.274** | **0.300** | 0.265 | 0.328 | **0.256** | **0.313** |
| | 720 | 1.059 | 0.741 | **0.332** | **0.372** | 0.411 | 0.393 | **0.351** | **0.353** | 0.376 | 0.397 | **0.329** | **0.366** |
| | Avg | 0.633 | 0.548 | **0.253** | **0.312** | 0.299 | 0.317 | **0.252** | **0.281** | 0.252 | 0.311 | **0.234** | **0.292** |
| | Pro | - | - | 60.0% | 43.0% | - | - | 15.7% | 11.3% | - | - | 7.1% | 6.1% |
| Traffic | 96 | 0.719 | 0.391 | **0.505** | **0.348** | 0.599 | 0.332 | **0.459** | **0.311** | 0.609 | 0.362 | **0.529** | **0.334** |
| | 192 | 0.696 | 0.379 | **0.521** | **0.354** | 0.619 | 0.341 | **0.475** | **0.315** | 0.623 | 0.365 | **0.519** | **0.327** |
| | 336 | 0.777 | 0.420 | **0.520** | **0.337** | 0.651 | 0.347 | **0.486** | **0.319** | 0.649 | 0.370 | **0.521** | **0.337** |
| | 720 | 0.864 | 0.472 | **0.552** | **0.352** | 0.658 | 0.358 | **0.522** | **0.338** | 0.758 | 0.418 | **0.556** | **0.350** |
| | Avg | 0.764 | 0.415 | **0.524** | **0.347** | 0.631 | 0.344 | **0.485** | **0.320** | 0.659 | 0.378 | **0.531** | **0.337** |
| | Pro | - | - | 31.4% | 16.3% | - | - | 23.1% | 6.9% | - | - | 19.4% | 10.8% |
| Electricity | 96 | 0.274 | 0.368 | **0.195** | **0.300** | 0.168 | 0.271 | **0.154** | **0.256** | 0.170 | 0.279 | **0.150** | **0.248** |
| | 192 | 0.296 | 0.386 | **0.193** | **0.291** | 0.186 | 0.285 | **0.163** | **0.263** | 0.198 | 0.303 | **0.168** | **0.263** |
| | 336 | 0.300 | 0.394 | **0.206** | **0.300** | 0.194 | 0.297 | **0.178** | **0.279** | 0.235 | 0.328 | **0.200** | **0.290** |
| | 720 | 0.373 | 0.439 | **0.241** | **0.332** | 0.224 | 0.316 | **0.201** | **0.299** | 0.270 | 0.360 | **0.235** | **0.323** |
| | Avg | 0.310 | 0.397 | **0.208** | **0.305** | 0.193 | 0.292 | 0.174 | **0.274** | 0.218 | 0.317 | **0.188** | **0.281** |
| | Pro | - | - | 32.9% | 23.1% | - | - | 9.8% | 6.1% | - | - | 13.7% | 11.3% |

We also list the forecasting results of Channel-independence baselines in Table 2. It is worth noting that InfoTime also outperforms Channel-independence baselines, indicating that although Channel-independence models exhibit promising results, incorporating cross-variable features can further enhance their effectiveness. Additionally, we evaluate InfoTime on the PEMS datasets, which consist of variables with clear geographical correlations. The results in Table 3 demonstrate a significant performance gap between PatchTST and RMLP in comparison to Informer, suggesting that Channel-independence models may not be optimal in scenarios where there are clear correlations between variables. In contrast, our framework exhibits improved performance for both Channel-mixing and Channel-independence models (We also verify the effectiveness of InfoTime on synthetic data, as show in Appendix A.2).

Table 2: Multivariate long-term series forecasting results on Channel-Independent models with different prediction lengths. We set the input length $I$ as 336 for all the models. The best result is indicated in bold font. See Table 6 in the Appendix for the full results.

| Models | | Metric | ETTh1 | | | | ETTm1 | | | | Traffic | | | |
|---|---|---|---|---|---|---|---|---|---|---|---|---|---|---|
| | | | 96 | 192 | 336 | 720 | 96 | 192 | 336 | 720 | 96 | 192 | 336 | 720 |
| PatchTST | Original | MSE | 0.375 | 0.414 | 0.440 | 0.460 | 0.290 | 0.332 | 0.366 | 0.420 | 0.367 | 0.385 | 0.398 | 0.434 |
| | | MAE | 0.399 | 0.421 | 0.440 | 0.473 | 0.342 | 0.369 | 0.392 | 0.424 | 0.251 | 0.259 | 0.265 | 0.287 |
| | w/Ours | MSE | **0.365** | **0.403** | **0.427** | **0.433** | **0.283** | **0.322** | **0.356** | **0.407** | **0.358** | **0.379** | **0.391** | **0.425** |
| | | MAE | **0.389** | **0.413** | **0.428** | **0.453** | **0.335** | **0.359** | **0.382** | **0.417** | **0.245** | **0.254** | **0.261** | **0.280** |
| RMLP | Original | MSE | 0.380 | 0.414 | 0.439 | 0.470 | 0.290 | 0.329 | 0.364 | 0.430 | 0.383 | 0.401 | 0.414 | 0.443 |
| | | MAE | 0.401 | 0.421 | 0.436 | 0.471 | 0.343 | 0.368 | 0.390 | 0.426 | 0.269 | 0.276 | 0.282 | 0.309 |
| | w/Ours | MSE | **0.367** | **0.404** | **0.426** | **0.439** | **0.285** | **0.322** | **0.358** | **0.414** | **0.364** | **0.384** | **0.398** | **0.428** |
| | | MAE | **0.391** | **0.413** | **0.429** | **0.459** | **0.335** | **0.359** | **0.381** | **0.413** | **0.249** | **0.258** | **0.266** | **0.284** |

Table 3: Multivariate long-term series forecasting results on three baselines and PEMS datasets with different prediction lengths. We set the input length $I$ as 336 for all the models. The best result is indicated in bold font. (See Table 9 for the ablation results of PEMS datasets.)

| Models | | Metric | PEMS03 | | | | PEMS04 | | | | PEMS08 | | | |
|---|---|---|---|---|---|---|---|---|---|---|---|---|---|---|
| | | | 96 | 192 | 336 | 720 | 96 | 192 | 336 | 720 | 96 | 192 | 336 | 720 |
| PatchTST | Original | MSE | 0.180 | 0.207 | 0.223 | 0.291 | 0.195 | 0.218 | 0.237 | 0.321 | 0.239 | 0.292 | 0.314 | 0.372 |
| | | MAE | 0.281 | 0.295 | 0.309 | 0.364 | 0.296 | 0.314 | 0.329 | 0.394 | 0.324 | 0.351 | 0.374 | 0.425 |
| | w/Ours | MSE | **0.115** | **0.154** | **0.164** | **0.198** | **0.110** | **0.118** | **0.129** | **0.149** | **0.114** | **0.160** | **0.177** | **0.209** |
| | | MAE | **0.223** | **0.251** | **0.256** | **0.286** | **0.221** | **0.224** | **0.237** | **0.261** | **0.218** | **0.243** | **0.241** | **0.281** |
| RMLP | Original | MSE | 0.160 | 0.184 | 0.201 | 0.254 | 0.175 | 0.199 | 0.210 | 0.255 | 0.194 | 0.251 | 0.274 | 0.306 |
| | | MAE | 0.257 | 0.277 | 0.291 | 0.337 | 0.278 | 0.294 | 0.306 | 0.348 | 0.279 | 0.311 | 0.328 | 0.365 |
| | w/Ours | MSE | **0.117** | **0.159** | **0.146** | **0.204** | **0.103** | **0.114** | **0.130** | **0.154** | **0.116** | **0.156** | **0.175** | **0.181** |
| | | MAE | **0.228** | **0.252** | **0.246** | **0.285** | **0.211** | **0.219** | **0.236** | **0.264** | **0.215** | **0.235** | **0.242** | **0.255** |
| Informer | Original | MSE | 0.139 | 0.152 | 0.165 | 0.216 | 0.132 | 0.146 | 0.147 | 0.145 | 0.156 | 0.175 | 0.187 | 0.264 |
| | | MAE | 0.240 | 0.252 | 0.260 | 0.290 | 0.238 | 0.249 | 0.247 | 0.245 | 0.262 | 0.266 | 0.274 | 0.325 |
| | w/Ours | MSE | **0.109** | **0.120** | **0.144** | **0.194** | **0.107** | **0.124** | **0.124** | **0.136** | **0.099** | **0.123** | **0.147** | **0.196** |
| | | MAE | **0.216** | **0.228** | **0.247** | **0.282** | **0.215** | **0.230** | **0.231** | **0.245** | **0.204** | **0.224** | **0.242** | **0.278** |
| Stationary | Original | MSE | 0.120 | 0.143 | 0.156 | 0.220 | 0.109 | 0.116 | 0.129 | 0.139 | 0.151 | 0.180 | 0.252 | 0.223 |
| | | MAE | 0.222 | 0.242 | 0.252 | 0.300 | 0.214 | 0.220 | 0.230 | 0.240 | 0.235 | 0.247 | 0.262 | 0.285 |
| | w/Ours | MSE | **0.101** | **0.131** | **0.153** | **0.190** | **0.096** | **0.114** | **0.125** | **0.135** | **0.103** | **0.144** | **0.184** | **0.217** |
| | | MAE | **0.206** | **0.229** | **0.245** | **0.273** | **0.199** | **0.217** | **0.229** | **0.243** | **0.200** | **0.220** | **0.245** | **0.278** |
| Crossformer | Original | MSE | 0.159 | 0.233 | 0.275 | 0.315 | 0.149 | 0.216 | 0.230 | 0.276 | 0.141 | 0.162 | 0.199 | 0.261 |
| | | MAE | 0.270 | 0.319 | 0.351 | 0.383 | 0.261 | 0.320 | 0.324 | 0.369 | 0.253 | 0.269 | 0.306 | 0.355 |
| | w/Ours | MSE | **0.119** | **0.166** | **0.189** | **0.223** | **0.114** | **0.139** | **0.161** | **0.171** | **0.088** | **0.108** | **0.134** | **0.171** |
| | | MAE | **0.217** | **0.250** | **0.265** | **0.293** | **0.215** | **0.236** | **0.258** | **0.275** | **0.190** | **0.206** | **0.222** | **0.251** |

## 4.2 ABLATION STUDY

In our approach, there are two components: CDAM and TAM. We perform an ablation study on the ETTh1, ETTh2, and Weather datasets with Informer and PatchTST. **+TAM** means that we add TAM to these baselines and **+InfoTime** means that we add both CDAM and TAM to baselines. We analyze the results shown in Table 4. Compared with baselines using the single-step forecaster, TAM performs better in most settings, which indicates the importance of cross-time correlation. For Channel-mixing models, we find that InfoTime can improve the performance of Channel-mixing models significantly, and alleviate the overfitting problem effectively. For Channel-Independence models, InfoTime can still improve the performance of Channel-Independence models, which indicates that correctly establishing the dependency between variables is an effective way to improve performance.

Table 4: Component ablation of InfoTime. We set the input length $I$ as 336 for PatchTST and 96 for Informer. The best results are in **bold** and the second best are underlined. (See Table 8 and Table 7 in the Appendix for the full ablation results.)

| Models | | Informer | | | | | | PatchTST | | | | | |
|---|---|---|---|---|---|---|---|---|---|---|---|---|---|
| | | Original | | +TAM | | +InfoTime | | Original | | +TAM | | +InfoTime | |
| Metric | | MSE | MAE | MSE | MAE | MSE | MAE | MSE | MAE | MSE | MAE | MSE | MAE |
| ETTh1 | 96 | 0.865 | 0.713 | 0.598 | 0.565 | **0.381** | **0.394** | 0.375 | 0.399 | 0.367 | 0.391 | **0.365** | **0.389** |
| | 192 | 1.008 | 0.792 | 0.694 | 0.640 | **0.435** | **0.430** | 0.414 | 0.421 | 0.405 | 0.414 | **0.403** | **0.413** |
| | 336 | 1.107 | 0.809 | 0.853 | 0.719 | **0.485** | **0.461** | 0.440 | 0.440 | 0.429 | 0.430 | **0.427** | **0.428** |
| | 720 | 1.181 | 0.865 | 0.914 | 0.741 | **0.534** | **0.524** | 0.460 | 0.473 | 0.435 | 0.455 | **0.433** | **0.453** |
| Weather | 96 | 0.300 | 0.384 | 0.277 | 0.354 | **0.179** | **0.249** | 0.152 | 0.199 | 0.149 | 0.197 | **0.144** | **0.194** |
| | 192 | 0.598 | 0.544 | 0.407 | 0.447 | **0.226** | **0.296** | 0.197 | 0.243 | 0.192 | 0.238 | **0.189** | **0.238** |
| | 336 | 0.578 | 0.523 | 0.529 | 0.520 | **0.276** | **0.334** | 0.250 | 0.284 | 0.247 | 0.280 | **0.239** | **0.279** |
| | 720 | 1.059 | 0.741 | 0.951 | 0.734 | **0.332** | **0.372** | 0.320 | 0.335 | 0.321 | 0.332 | **0.312** | **0.331** |

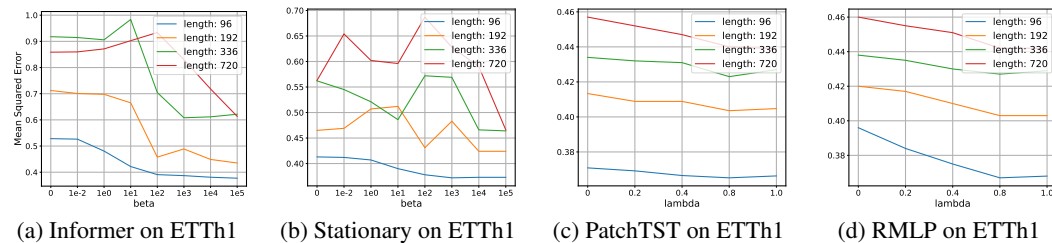

   (a) Informer on ETTh1   (b) Stationary on ETTh1   (c) PatchTST on ETTh1   (d) RMLP on ETTh1

Figure 4: Evaluation on hyper-parameter $\beta$ and $\lambda$. We evaluate the impact of $\beta$ with Informer and Stationary on the ETTh1 dataset, we also evaluate $\lambda$ with PatchTST and RMLP on the ETTh1 dataset.

### 4.3 EFFECT OF HYPER-PARAMETERS

We evaluate the effect of hyper-parameter $\beta$ on the ETTh1 and ETTm2 datasets with two baselines. In Figure 4, we increase the value of $\beta$ from 0 to $1e^5$ and evaluate MSE with different prediction windows on two datasets and two baselines. When $\beta$ is small, baselines perform poorly and unstably As $\beta$ increases, baselines perform better and more stable. In addition, as the prediction window increases, the overfitting problem of baselines is more and more serious, so a larger $\beta$ is needed to remove superfluous information. We also evaluate $\lambda$ with PatchTST and RMLP, we observe that the larger the $\lambda$, the better models' performance, and when $\lambda \geq 0.8$, the performance is stable.

## 5 CONCLUSION

This paper investigates two key factors in MTSF: temporal correlation and cross-variable correlation. To utilize the cross-variable correlation while eliminating the superfluous information, we introduce **C**ross-Variable **D**ecorrelation **A**ware **M**odeling (CDAM). In addition, we also propose **T**emporal **C**orrelation **A**ware **M**odeling (TAM) to model temporal correlations of predicted series. Integrating CDAM and TAM, we build a novel time series modeling framework for MTSF termed . Extensive experiments on various real-world MTSF datasets demonstrate the effectiveness of our framework.

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

## A  APPENDIX

### A.1  BASELINES

We choose SOTA and the most representative LTSF models as our baselines, including Channel-Independence models and Channel-mixing models.

- PatchTST (Nie et al., 2022): the current SOTA LTSF models. It utilizes channel-independence and patch techniques and achieves the highest performance by utilizing the native Transformer. We directly use the public official source code.[4]
- Informer (Zhou et al., 2021): it proposes improvements to the Transformer model by utilizing the a sparse self-attention mechanism. We take the official open source code.[5]
- NSformer Liu et al. (2022b): NSformer is to address the over-stationary problem and it devises the De-stationary Attention to recover the intrinsic non-stationay information into temporal dependencies by approximating distinguishable attentions learned from raw series. We also take the official implemented code.[6]
- Crossformer Zhang & Yan: similar to PatchTST, it also utilizes the patch techniques. Unlike PatchTST, it leverages cross-variable and cross-time attention. We utilize the official code. [7] and fix the input length to 96.
- RMLP: it if a linear-based models which consists of two linear layers with relu activation.

For the ETT, Weather, Electricity, and traffic datasets, we set $I = 96$ for Channel-mixing models and $I = 336$ for Channel-Independence models, as longer input lengths tend to yield better performance for Channel-Independence models. For PEMS03, PEMS04, and PEMS08 datasets, we set $I = 336$ for all of these models since all of them perform better in a longer input length.

### A.2  SYNTHETIC DATA

To demonstrate that InfoTime can take advantages of cross-variable correlation while avoiding unnecessary noise, we also conducted experiments on simulated data. The function for the synthetic data is $y_i = \sum_{j=1}^{J} A_i^j sin(\omega_i^j x + \varphi_i^j)$ for $x_i \in [0, 1]$, where the frequencies, amplitude and phase shifts are randomly selected via $\omega_i^j \sim \mathcal{U}(0, \pi)$, $A_i^j \sim \mathcal{U}(0, 1)$, $\varphi_i^j \sim \mathcal{U}(0, \pi)$, and we set $J = 5$. Meanwhile, we add Gaussian noise $\epsilon \sim \mathcal{N}(0, \sigma)$ to simulate the noise situation, which is independent of $y$. And the experimental settings are shown in Table 5.

---

[4]https://github.com/yuqinie98/PatchTST

[5]https://github.com/zhouhaoyi/Informer2020

[6]https://github.com/thuml/Nonstationary_Transformers

[7]https://github.com/Thinklab-SJTU/Crossformer

Table 5: Experimental settings on Synthetic Data.

| | Channel-mixing | Channel-Independence | CDAM |
|---|---|---|---|
| Input | $i_t = y_{t-T:t}$ 
 Amplitude $A_{t+1:t+P}$ 
 Frequency $\omega_{t+1:t+P}$ 
 Phase Shifts $\varphi_{t+1:t+P}$ 
 Variable $x_{t+1:t+P}$ 
 Noise $\epsilon$ | $i_t = y_{t-T:t}$ | $i_t = y_{t-T:t}$ 
 $A_{t+1:t+P}$ 
 Frequency $\omega_{t+1:t+P}$ 
 Phase Shifts $\varphi_{t+1:t+P}$ 
 Variable $x_{t+1:t+P}$ 
 Noise $\epsilon$ |
| Output | $o_t = y_{t+1:t+P}$ | $o_t = y_{t+1:t+P}$ | $o_t = y_{t+1:t+P}$ |
| Time Mixing | 3-layers MLP | 3-layers MLP | - |
| Channel Mixing | 2-layers MLP | - | Directly add to CM Model |

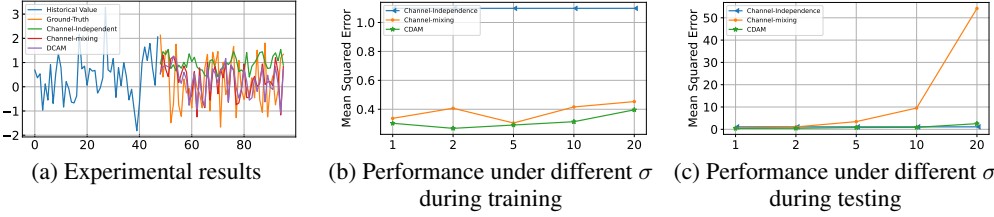

(a) Experimental results    (b) Performance under different $\sigma$ during training    (c) Performance under different $\sigma$ during testing

Figure 5: Experimental results on synthetic data.

The experimental results on synthetic data are presented in Figure 5. Since the Channel-Independence model only takes into account the historical value and variables such as $A, \omega, \varphi, x$ change over time and are not fixed, its performance is notably inferior to that of the Channel-mixing model and CDAM. Additionally, we conducted experiments to investigate the impact of noise by training these three models using different noise levels, manipulated through the adjustment of $\sigma$, while maintaining a consistent $\sigma$ during testing. The experimental results are shown in Figure 5 (b). We observed that when the noise in the training set and the test set follows the same distribution, noise has a minimal effect on the models' performance, resulting in only a slight reduction. Notably, compared to the Channel-mixing model, our CDAM model performs consistently well across various noise levels. To further demonstrate the noise reduction capability of CDAM, we set $\sigma = 1$ during training and modified the variance $\sigma$ during testing. As depicted in Figure 5 (c), we observed that the performance of CDAM remains relatively stable, while the effectiveness of the Channel-mixing model is significantly impacted. This observation highlights the ability of CDAM to effectively minimize the influence of irrelevant noise, despite the fact that complete elimination is not achieved.

## A.3 LOWER BOUND FOR $I(X^i, Y^i; Z^i)$

### A.3.1 DERIVATION OF $I(X^i, Y^i; Z^i)$

The Mutual Information between the i-th historical series $X^i$, i-th future series $Y^i$ and latent representation $Z^i$ is defined as:

$$
\begin{aligned}
I(X^i, Y^i; Z^i) &= I(X^i; Z^i) + I(Z^i; Y^i | X^i) \\
&= \int p(z^i, x^i) \log \frac{p(z^i, x^i)}{p(z^i)p(x^i)} dz^i dx^i + \int p(z^i, y^i, x^i) \log \frac{p(x^i)p(z^i, y^i, x^i)}{p(z^i, x^i)p(y^i, x^i)} dz^i dy^i dx^i \\
&= \int p(z^i, y^i, x^i) \log \frac{p(z^i, x^i)}{p(z^i)p(x^i)} dz^i dy^i dx^i + \int p(z^i, y^i, x^i) \log \frac{p(x^i)p(z^i, y^i, x^i)}{p(z^i, x^i)p(y^i, x^i)} dz^i dy^i dx^i \\
&= \int p(z^i, y^i, x^i) \log \frac{p(z^i, y^i, x^i)}{p(z^i)p(y^i, x^i)} dz^i dy^i dx^i \\
&= \int p(z^i, y^i, x^i) \log \frac{p(x^i, y^i | z^i)}{p(y^i, x^i)} dz^i dy^i dx^i \\
&= \int p(z^i, y^i, x^i) \log p(y^i, x^i | z^i) dz^i dy^i dx^i - \int p(z^i, y^i, x^i) \log p(y^i, x^i) dz^i dy^i dx^i \\
&= \int p(z^i, y^i, x^i) \log p(y^i | x^i, z^i) p(x^i | z^i) dz^i dy^i dx^i - \int p(y^i, x^i) \log p(y^i, x^i) dy^i dx^i \\
&= \mathbb{E}_{p(z^i, y^i, x^i)} \left[ \log p(y^i | x^i, z^i) \right] + \mathbb{E}_{p(z^i, x^i)} \left[ \log p(x^i | z^i) \right] + H(Y^i, X^i)
\end{aligned}
\tag{14}
$$

Therefore, $I(X^i; Y^i; Z^i)$ can be represented as:

$$
I(X^i, Y^i | Z^i) = \mathbb{E}_{p(z^i, y^i, x^i)} \left[ \log p(y^i | x^i, z^i) \right] + \mathbb{E}_{p(z^i, x^i)} \left[ \log p(x^i | z^i) \right] + H(Y^i, X^i) \tag{15}
$$

### A.3.2 VARIATIONAL APPROXIMATION OF $I(X^i, Y^i; Z^i)$

$$
I(X^i, Y^i; Z^i) = \mathbb{E}_{p(z^i, y^i, x^i)} \left[ \log p(y^i | x^i, z^i) \right] + \mathbb{E}_{p(z^i, x^i)} \left[ \log p(x^i | z^i) \right] + \text{constant} \tag{16}
$$

We introduce $p_\theta(x^i | z^i)$ to be the variational approximation of $p(x^i, y^i)$. Since the Kullback Leibler (KL) divergence is always non-negetive, we have:

$$
\begin{aligned}
D_{KL} \left[ p(X^i | Z^i) || p_\theta(X^i | Z^i) \right] &= \int p(x^i | z^i) log \frac{p(x^i | z^i)}{p(x^i)p(z^i)} dx^i dz^i \geq 0 \\
\mathbb{E}_{p(z^i, x^i)} \left[ \log p(x^i | z^i) \right] &\geq \mathbb{E}_{p(z^i, x^i)} \left[ \log p_\theta(x^i | z^i) \right]
\end{aligned}
\tag{17}
$$

In the same way, we have:

$$
\mathbb{E}_{p(z^i, y^i, x^i)} \left[ \log p(y^i | x^i, z^i) \right] \geq \mathbb{E}_{p(z^i, y^i, x^i)} \left[ \log p_\theta(y^i | x^i, z^i) \right] \tag{18}
$$

Therefore, the variational lower bound is as follows:

$$
I(X^i, Y^i; Z^i) - constant \geq I_v(X^i, Y^i; Z^i) = \mathbb{E}_{p(z^i, y^i, x^i)} \left[ \log p_\theta(y^i | x^i, z^i) \right] + \mathbb{E}_{p(z^i, x^i)} \left[ \log p_\theta(x^i | z^i) \right] \tag{19}
$$

$I(X^i, Y^i; Z^i)$ can thus be maximized by maximizing its variational lower bound.

### A.4 DERIVATION AND VARIATIONAL APPROXIMATION OF $I(Y_j^i; Y_{j-1}^i | X^i)$

The Mutual Information between the $j$-th downsampling predicted subsequence $\hat{Y}_j^i$ and $(j-1)$-th target downsampling subsequence $Y_{j-1}^i$ given historical sequence $X^i$ is defined as:

$$
I(\hat{Y}_j^i; Y_{j-1}^i | X^i) = \int p(\hat{y}_j^i, y_{j-1}^i, x^i) \log \frac{p(\hat{y}_j^i, y_{j-1}^i | x^i)}{p(\hat{y}_j^i | x^i)p(y_{j-1}^i | x^i)} d\hat{y}_j^i dy_{j-1}^i dx^i \tag{20}
$$

$$
\begin{aligned}
I(Y_{j-1}^i; \hat{Y}_j^i | X^i) &= \int p(y_{j-1}^i, x^i, \hat{y}_j^i) \log \frac{p(x^i)p(y_{j-1}^i, \hat{y}_j^i, x^i)}{p(y_{j-1}^i, x^i)p(\hat{y}_j^i, x^i)} d\hat{y}_{j-1}^i dy_j^i dx^i \\
&= \int p(y_{j-1}^i, x^i, \hat{y}_j^i) \log \frac{p(y_{j-1}^i | \hat{y}_j^i, x^i)}{p(y_{j-1}^i | x^i)} d\hat{y}_{j-1}^i dy_j^i dx^i \\
&= \int p(y_{j-1}^i, x^i, \hat{y}_j^i) \log p(y_{j-1}^i | \hat{y}_j^i, x^i) dy_{j-1}^i d\hat{y}_j^i dx^i + H(Y_{j-1}^i, X^i) \\
&\geq \int p(y_{j-1}^i, x^i, \hat{y}_j^i) \log p(y_{j-1}^i | \hat{y}_j^i, x^i) dy_{j-1}^i d\hat{y}_j^i dx^i
\end{aligned}
\tag{21}
$$

Since $p(y_{j-1}^i | \hat{y}_j^i, x^i)$ is intractable, we use $p_\theta(y_{j-1}^i | \hat{y}_j^i, x^i)$ to approximate $p(\hat{y}_j^i | y_{j-1}^i, x^i)$, therefore, we have:

$$
\begin{aligned}
I(Y_{j-1}^i; \hat{Y}_j^i | X^i) &\geq \int p(y_{j-1}^i, x^i, \hat{y}_j^i) \log p(y_{j-1}^i | \hat{y}_j^i, x^i) dy_{j-1}^i d\hat{y}_j^i dx^i \\
&\geq \int p(y_{j-1}^i, \hat{y}_j^i, x^i) \log \frac{p(y_{j-1}^i | \hat{y}_j^i, x^i) p_\theta(y_{j-1}^i | \hat{y}_j^i, x^i)}{p_\theta(y_{j-1}^i | \hat{y}_j^i, x^i)} dy_{j-1}^i d\hat{y}_j^i dx^i \\
&\geq \int p(y_{j-1}^i, \hat{y}_j^i, x^i) \log p_\theta(y_{j-1}^i | \hat{y}_j^i, x^i) dy_{j-1}^i d\hat{y}_j^i dx^i \\
&\geq \mathbb{E}_{p(y_{j-1}^i, \hat{y}_j^i, x^i)} \left[ p_\theta(y_{j-1}^i | \hat{y}_j^i, x^i) \right]
\end{aligned}
\tag{22}
$$

Therefore, the Mutual information $I(Y_{j-1}^i; \hat{Y}_j^i | X^i)$ can be maximized by maximizing $\mathbb{E}_{p(y_{j-1}^i, \hat{y}_j^i, x^i)} \left[ p_\theta(y_{j-1}^i | \hat{y}_j^i, x^i) \right]$.

## A.5 EXTRA EXPERIMENTAL RESULTS

### A.5.1 FULL RESULTS OF CHANNEL-INDEPENDENCE MODELS

In this section, we provide the full experimental result of Channel-Independence models in Table 6 which is an extended version of Table 2

### A.5.2 ADDITIONAL ABLATION STUDY

In this section, we provide the full ablation experimental results of the Channel-mixing models and Channel-Independence models in Table 8 and Table 7, respectively, which are the extended of Table 4. We also provide Table 9, which contains the full results of ablation experiments on the PEMS (PEMS03, PEMS04, PEMS08) dataset.

Table 6: Multivariate long-term series forecasting results on Channel-Independence models with different prediction lengths $O \in \{96, 192, 336, 720\}$. We set the input length $I$ as 336 for all the models. The best result is indicated in bold font. (*Avg* is averaged from all four prediction lengths and *Pro* means the relative MSE and MAE reduction.)

| Models | | PatchTST | | | | RMLP | | | |
|---|---|---|---|---|---|---|---|---|---|
| | | Original | | w/Ours | | Original | | w/Ours | |
| Metric | | MSE | MAE | MSE | MAE | MSE | MAE | MSE | MAE |
| ETTh1 | 96 | 0.375 | 0.399 | **0.365** | **0.389** | 0.380 | 0.401 | **0.367** | **0.391** |
| | 192 | 0.414 | 0.421 | **0.403** | **0.413** | 0.414 | 0.421 | **0.404** | **0.413** |
| | 336 | 0.440 | 0.440 | **0.427** | **0.428** | 0.439 | 0.436 | **0.426** | **0.429** |
| | 720 | 0.460 | 0.473 | **0.433** | **0.453** | 0.470 | 0.471 | **0.439** | **0.459** |
| | Avg | 0.422 | 0.433 | **0.407** | **0.420** | 0.426 | 0.432 | **0.409** | **0.423** |
| | Pro | - | - | **3.5%** | **3.0%** | - | - | **3.9%** | **2.1%** |
| ETTh2 | 96 | 0.274 | 0.335 | **0.271** | **0.332** | 0.290 | 0.348 | **0.271** | **0.333** |
| | 192 | 0.342 | 0.382 | **0.334** | **0.373** | 0.350 | 0.388 | **0.335** | **0.374** |
| | 336 | 0.365 | 0.404 | **0.357** | **0.395** | 0.374 | 0.410 | **0.358** | **0.395** |
| | 720 | 0.391 | 0.428 | **0.385** | **0.421** | 0.410 | 0.439 | **0.398** | **0.432** |
| | Avg | 0.343 | 0.387 | **0.337** | **0.380** | 0.356 | 0.396 | **0.34** | **0.384** |
| | Pro | - | - | **1.7%** | **1.8%** | - | - | **4.5%** | **3.0%** |
| ETTm1 | 96 | 0.290 | 0.342 | **0.283** | **0.335** | 0.290 | 0.343 | **0.285** | **0.335** |
| | 192 | 0.332 | 0.369 | **0.322** | **0.359** | 0.329 | 0.368 | **0.322** | **0.359** |
| | 336 | 0.366 | 0.392 | **0.356** | **0.382** | 0.364 | 0.390 | **0.358** | **0.381** |
| | 720 | 0.420 | 0.424 | **0.407** | **0.417** | 0.430 | 0.426 | **0.414** | **0.413** |
| | Avg | 0.352 | 0.381 | **0.342** | **0.373** | 0.353 | 0.381 | **0.344** | **0.372** |
| | Pro | - | - | **2.8%** | **2.1%** | - | - | **2.5%** | **2.3%** |
| ETTm2 | 96 | 0.165 | 0.255 | **0.161** | **0.250** | 0.177 | 0.263 | **0.162** | **0.252** |
| | 192 | 0.220 | 0.292 | **0.217** | **0.289** | 0.233 | 0.302 | **0.217** | **0.289** |
| | 336 | 0.278 | 0.329 | **0.271** | **0.324** | 0.283 | 0.335 | **0.270** | **0.324** |
| | 720 | 0.367 | 0.385 | **0.362** | **0.381** | 0.367 | 0.388 | **0.357** | **0.380** |
| | Avg | 0.257 | 0.315 | **0.252** | **0.311** | 0.265 | 0.322 | **0.251** | **0.311** |
| | Pro | - | - | **1.9%** | **1.3%** | - | - | **5.2%** | **3.4%** |
| Weather | 96 | 0.152 | 0.199 | **0.144** | **0.194** | 0.147 | 0.198 | **0.144** | **0.196** |
| | 192 | 0.197 | 0.243 | **0.189** | **0.238** | 0.190 | 0.239 | **0.187** | **0.237** |
| | 336 | 0.250 | 0.284 | **0.239** | **0.279** | 0.243 | 0.280 | **0.239** | **0.277** |
| | 720 | 0.320 | 0.335 | **0.312** | **0.331** | 0.320 | 0.332 | **0.316** | **0.330** |
| | Avg | 0.229 | 0.265 | **0.221** | **0.260** | 0.225 | 0.262 | **0.221** | **0.260** |
| | Pro | - | - | **3.5%** | **1.8%** | - | - | **1.6%** | **0.9%** |
| Traffic | 96 | 0.367 | 0.251 | **0.358** | **0.245** | 0.383 | 0.269 | **0.364** | **0.249** |
| | 192 | 0.385 | 0.259 | **0.379** | **0.254** | 0.401 | 0.276 | **0.384** | **0.258** |
| | 336 | 0.398 | 0.265 | **0.391** | **0.261** | 0.414 | 0.282 | **0.398** | **0.266** |
| | 720 | 0.434 | 0.287 | **0.425** | **0.280** | 0.443 | 0.309 | **0.428** | **0.284** |
| | Avg | 0.396 | 0.265 | **0.388** | **0.260** | 0.410 | 0.284 | **0.393** | **0.264** |
| | Pro | - | - | **2.0%** | **1.8%** | - | - | **5.2%** | **8.4%** |
| Electricity | 96 | 0.130 | 0.222 | **0.125** | **0.219** | 0.130 | 0.225 | **0.125** | **0.218** |
| | 192 | 0.148 | 0.242 | **0.143** | **0.235** | 0.148 | 0.240 | **0.144** | **0.236** |
| | 336 | 0.167 | 0.261 | **0.161** | **0.255** | 0.164 | 0.257 | **0.160** | **0.253** |
| | 720 | 0.202 | 0.291 | **0.198** | **0.287** | 0.203 | 0.291 | **0.195** | **0.285** |
| | Avg | 0.161 | 0.254 | **0.156** | **0.249** | 0.161 | 0.253 | **0.156** | **0.248** |
| | Pro | - | - | **3.1%** | **1.9%** | - | - | **3.1%** | **2.1%** |

Table 7: Component ablation of InfoTime for RMLP and PatchTST. We set the input length $I$ as 336. The best results are in **bold** and the second best are underlined.

| Models | | RMLP | | | | | | PatchTST | | | | | |
|---|---|---|---|---|---|---|---|---|---|---|---|---|---|
| | | Original | | +TAM | | +InfoTime | | Original | | +TAM | | +InfoTime | |
| Metric | | MSE | MAE | MSE | MAE | MSE | MAE | MSE | MAE | MSE | MAE | MSE | MAE |
| ETTh1 | 96 | 0.380 | 0.401 | _0.371_ | _0.392_ | **0.367** | **0.391** | 0.375 | 0.399 | _0.367_ | _0.391_ | **0.365** | **0.389** |
| | 192 | 0.414 | 0.421 | _0.406_ | _0.414_ | **0.404** | **0.413** | 0.414 | 0.421 | _0.405_ | _0.414_ | **0.403** | **0.413** |
| | 336 | 0.439 | 0.436 | _0.427_ | **0.428** | **0.426** | _0.429_ | 0.440 | 0.440 | _0.429_ | _0.430_ | **0.427** | **0.428** |
| | 720 | 0.470 | 0.471 | 0.450 | 0.465 | **0.439** | **0.459** | 0.460 | 0.473 | _0.435_ | _0.455_ | **0.433** | **0.453** |
| ETTh2 | 96 | 0.290 | 0.348 | _0.278_ | _0.337_ | **0.271** | **0.333** | 0.274 | 0.335 | **0.271** | **0.332** | 0.271 | 0.332 |
| | 192 | 0.350 | 0.388 | _0.340_ | _0.377_ | **0.335** | **0.374** | 0.342 | 0.382 | 0.334 | **0.373** | 0.334 | 0.373 |
| | 336 | 0.374 | 0.410 | _0.366_ | _0.402_ | **0.358** | _0.395_ | 0.365 | 0.404 | **0.357** | **0.393** | 0.357 | _0.395_ |
| | 720 | 0.410 | 0.439 | _0.404_ | _0.435_ | **0.398** | **0.432** | 0.391 | 0.428 | _0.386_ | _0.422_ | **0.385** | **0.421** |
| ETTm1 | 96 | 0.290 | 0.343 | **0.285** | _0.337_ | **0.285** | **0.335** | 0.290 | 0.342 | _0.286_ | _0.337_ | **0.283** | **0.335** |
| | 192 | 0.329 | 0.368 | **0.321** | _0.360_ | _0.322_ | **0.359** | 0.332 | 0.369 | _0.326_ | _0.366_ | **0.322** | **0.359** |
| | 336 | 0.364 | 0.390 | **0.357** | _0.382_ | _0.358_ | **0.381** | 0.366 | 0.392 | _0.359_ | _0.388_ | **0.356** | **0.382** |
| | 720 | 0.430 | 0.426 | _0.415_ | _0.415_ | **0.414** | **0.413** | 0.420 | 0.424 | _0.408_ | **0.413** | 0.407 | _0.417_ |
| ETTm2 | 96 | 0.177 | 0.263 | _0.166_ | _0.255_ | **0.162** | **0.252** | 0.165 | 0.255 | _0.162_ | _0.251_ | **0.161** | **0.250** |
| | 192 | 0.233 | 0.302 | _0.222_ | _0.294_ | **0.217** | **0.289** | 0.220 | 0.292 | _0.218_ | _0.290_ | **0.217** | **0.289** |
| | 336 | 0.283 | 0.335 | _0.274_ | _0.328_ | **0.270** | **0.324** | 0.278 | 0.329 | _0.273_ | _0.326_ | **0.271** | **0.324** |
| | 720 | 0.367 | 0.388 | _0.362_ | _0.384_ | **0.357** | **0.380** | 0.367 | 0.385 | _0.364_ | _0.382_ | **0.362** | **0.381** |
| Weather | 96 | 0.147 | 0.198 | _0.146_ | _0.197_ | **0.144** | **0.196** | 0.152 | 0.199 | _0.149_ | _0.197_ | **0.144** | **0.194** |
| | 192 | 0.190 | 0.239 | _0.189_ | _0.238_ | **0.187** | **0.237** | 0.197 | 0.243 | _0.192_ | **0.238** | 0.189 | 0.238 |
| | 336 | 0.243 | 0.280 | _0.241_ | _0.278_ | **0.239** | **0.277** | 0.250 | 0.284 | _0.247_ | _0.280_ | **0.239** | **0.279** |
| | 720 | _0.320_ | 0.332 | 0.319 | _0.332_ | **0.316** | **0.330** | 0.320 | 0.335 | 0.321 | _0.332_ | **0.312** | **0.331** |
| Traffic | 96 | 0.383 | 0.269 | _0.366_ | _0.252_ | **0.364** | **0.249** | 0.367 | 0.251 | _0.359_ | **0.245** | 0.358 | **0.245** |
| | 192 | 0.401 | 0.276 | _0.386_ | _0.260_ | **0.384** | **0.258** | 0.385 | 0.259 | _0.380_ | _0.255_ | **0.379** | **0.254** |
| | 336 | 0.414 | 0.282 | _0.400_ | _0.268_ | **0.398** | **0.266** | 0.398 | 0.265 | **0.391** | _0.262_ | **0.391** | **0.261** |
| | 720 | 0.443 | 0.309 | _0.432_ | _0.286_ | **0.428** | **0.284** | 0.434 | 0.287 | **0.424** | **0.279** | _0.425_ | _0.280_ |
| Electricity | 96 | 0.130 | 0.225 | 0.127 | 0.221 | **0.125** | **0.218** | 0.130 | _0.222_ | _0.129_ | 0.223 | **0.125** | **0.219** |
| | 192 | 0.148 | 0.240 | 0.145 | 0.238 | **0.144** | **0.236** | 0.148 | 0.242 | _0.147_ | _0.240_ | **0.143** | **0.235** |
| | 336 | 0.164 | 0.257 | 0.162 | 0.255 | **0.160** | **0.253** | 0.167 | 0.261 | _0.165_ | _0.258_ | **0.161** | **0.255** |
| | 720 | 0.203 | 0.291 | 0.199 | 0.288 | **0.195** | **0.285** | _0.202_ | _0.291_ | 0.204 | 0.292 | **0.198** | **0.287** |

Table 8: Component ablation of InfoTime for Informer, Stationary, and Crossformer. We set the input length $I$ as 96. The best results are in **bold** and the second best are underlined.

| Models | | Informer | | | | | | Stationary | | | | | | Crossformer | | | | | |
|---|---|---|---|---|---|---|---|---|---|---|---|---|---|---|---|---|---|---|---|
| | | Original | | +TAM | | +InfoTime | | Original | | +TAM | | +InfoTime | | Original | | +TAM | | +InfoTime | |
| | Metric | MSE | MAE | MSE | MAE | MSE | MAE | MSE | MAE | MSE | MAE | MSE | MAE | MSE | MAE | MSE | MAE | MSE | MAE |
| ETTh1 | 96 | 0.865 | 0.713 | 0.598 | 0.565 | **0.381** | **0.394** | 0.598 | 0.498 | 0.455 | 0.452 | **0.375** | **0.388** | 0.457 | 0.463 | 0.396 | 0.411 | **0.379** | **0.392** |
| | 192 | 1.008 | 0.792 | 0.694 | 0.640 | **0.435** | **0.430** | 0.602 | 0.520 | 0.491 | 0.478 | **0.425** | **0.417** | 0.635 | 0.581 | 0.541 | 0.511 | **0.433** | **0.427** |
| | 336 | 1.107 | 0.809 | 0.853 | 0.719 | **0.485** | **0.461** | 0.677 | 0.573 | 0.611 | 0.530 | **0.463** | **0.436** | 0.776 | 0.667 | 0.759 | 0.651 | **0.482** | **0.458** |
| | 720 | 1.181 | 0.865 | 0.914 | 0.741 | **0.534** | **0.524** | 0.719 | 0.597 | 0.594 | 0.542 | **0.463** | **0.459** | 0.861 | 0.725 | 0.845 | 0.711 | **0.529** | **0.517** |
| ETTh2 | 96 | 3.755 | 1.525 | 0.502 | 0.538 | **0.336** | **0.390** | 0.362 | 0.393 | 0.330 | 0.371 | **0.286** | **0.335** | 0.728 | 0.615 | 0.364 | 0.415 | **0.333** | **0.386** |
| | 192 | 5.602 | 1.931 | 0.821 | 0.701 | **0.468** | **0.470** | 0.481 | 0.453 | 0.456 | 0.440 | **0.371** | **0.388** | 0.898 | 0.705 | 0.470 | 0.481 | **0.455** | **0.453** |
| | 336 | 4.721 | 1.835 | 1.065 | 0.823 | **0.582** | **0.534** | 0.524 | 0.487 | 0.475 | 0.463 | **0.414** | **0.425** | 1.132 | 0.807 | 0.580 | 0.547 | **0.554** | **0.513** |
| | 720 | 3.647 | 1.625 | 1.489 | 1.022 | **0.749** | **0.620** | 0.512 | 0.494 | 0.506 | 0.486 | **0.418** | **0.437** | 4.390 | 1.795 | 0.768 | 0.648 | **0.757** | **0.619** |
| ETTm1 | 96 | 0.672 | 0.571 | 0.435 | 0.444 | **0.326** | **0.367** | 0.396 | 0.401 | 0.375 | 0.396 | **0.326** | **0.362** | 0.385 | 0.409 | 0.388 | 0.401 | **0.323** | **0.362** |
| | 192 | 0.795 | 0.669 | 0.473 | 0.467 | **0.371** | **0.391** | 0.471 | 0.436 | 0.441 | 0.432 | **0.366** | **0.379** | 0.459 | 0.478 | 0.436 | 0.428 | **0.366** | **0.386** |
| | 336 | 1.212 | 0.871 | 0.545 | 0.518 | **0.408** | **0.416** | 0.517 | 0.464 | 0.472 | 0.455 | **0.392** | **0.398** | 0.645 | 0.583 | 0.483 | 0.457 | **0.403** | **0.414** |
| | 720 | 1.166 | 0.823 | 0.669 | 0.589 | **0.482** | **0.464** | 0.664 | 0.527 | 0.532 | 0.489 | **0.455** | **0.434** | 0.756 | 0.669 | 0.548 | 0.498 | **0.473** | **0.460** |
| ETTm2 | 96 | 0.365 | 0.453 | 0.258 | 0.378 | **0.187** | **0.282** | 0.201 | 0.291 | 0.185 | 0.276 | **0.175** | **0.256** | 0.281 | 0.373 | 0.223 | 0.321 | **0.186** | **0.281** |
| | 192 | 0.533 | 0.563 | 0.439 | 0.515 | **0.277** | **0.351** | 0.275 | 0.335 | 0.254 | 0.318 | **0.238** | **0.297** | 0.549 | 0.520 | 0.347 | 0.404 | **0.269** | **0.341** |
| | 336 | 1.363 | 0.887 | 0.836 | 0.728 | **0.380** | **0.420** | 0.350 | 0.377 | 0.343 | 0.372 | **0.299** | **0.336** | 0.729 | 0.603 | 0.528 | 0.506 | **0.356** | **0.396** |
| | 720 | 3.379 | 1.338 | 3.172 | 1.322 | **0.607** | **0.549** | 0.460 | 0.435 | 0.440 | 0.421 | **0.398** | **0.393** | 1.059 | 0.741 | 0.895 | 0.665 | **0.493** | **0.482** |
| Weather | 96 | 0.300 | 0.384 | 0.277 | 0.354 | **0.179** | **0.249** | 0.181 | 0.230 | 0.178 | 0.226 | **0.166** | **0.213** | 0.158 | 0.236 | 0.154 | 0.225 | **0.149** | **0.218** |
| | 192 | 0.598 | 0.544 | 0.407 | 0.447 | **0.226** | **0.296** | 0.286 | 0.312 | 0.261 | 0.296 | **0.218** | **0.260** | 0.209 | 0.285 | 0.202 | **0.27** | **0.202** | 0.272 |
| | 336 | 0.578 | 0.523 | 0.529 | 0.520 | **0.276** | **0.334** | 0.319 | 0.335 | 0.318 | 0.333 | **0.274** | **0.300** | 0.265 | 0.328 | 0.263 | 0.320 | **0.256** | **0.313** |
| | 720 | 1.059 | 0.741 | 0.951 | 0.734 | **0.332** | **0.372** | 0.411 | 0.393 | 0.387 | 0.378 | **0.351** | **0.353** | 0.376 | 0.397 | 0.353 | 0.382 | **0.329** | **0.366** |
| Traffic | 96 | 0.719 | 0.391 | 0.577 | 0.356 | **0.505** | **0.348** | 0.599 | 0.332 | 0.503 | 0.313 | **0.459** | **0.311** | 0.609 | 0.362 | **0.490** | **0.308** | 0.529 | 0.334 |
| | 192 | 0.696 | 0.379 | 0.556 | 0.357 | **0.521** | **0.354** | 0.619 | 0.341 | 0.488 | 0.309 | **0.475** | **0.315** | 0.623 | 0.365 | **0.493** | **0.310** | 0.519 | 0.327 |
| | 336 | 0.777 | 0.420 | 0.580 | 0.370 | **0.520** | **0.337** | 0.651 | 0.347 | 0.506 | 0.318 | **0.486** | **0.319** | 0.649 | 0.370 | 0.53 | 0.328 | **0.521** | **0.337** |
| | 720 | 0.864 | 0.472 | 0.668 | 0.430 | **0.552** | **0.352** | 0.658 | 0.358 | 0.542 | 0.329 | **0.522** | **0.338** | 0.758 | 0.418 | 0.591 | 0.348 | **0.556** | **0.350** |
| Electricity | 96 | 0.274 | 0.368 | 0.228 | 0.333 | **0.195** | **0.300** | 0.168 | 0.271 | **0.152** | **0.252** | 0.154 | 0.256 | 0.170 | 0.279 | 0.151 | 0.251 | **0.150** | **0.248** |
| | 192 | 0.296 | 0.386 | 0.238 | 0.344 | **0.193** | **0.291** | 0.186 | 0.285 | 0.166 | 0.265 | **0.163** | **0.263** | 0.198 | 0.303 | **0.168** | 0.266 | 0.168 | **0.263** |
| | 336 | 0.300 | 0.394 | 0.254 | 0.358 | **0.206** | **0.300** | 0.194 | 0.297 | 0.180 | 0.280 | **0.178** | **0.279** | 0.235 | 0.328 | **0.197** | 0.292 | 0.200 | **0.290** |
| | 720 | 0.373 | 0.439 | 0.288 | 0.379 | **0.241** | **0.332** | 0.224 | 0.316 | 0.208 | 0.305 | **0.201** | **0.299** | 0.27 | 0.36 | 0.238 | 0.328 | **0.235** | **0.323** |

Table 9: Component ablation of InfoTime for PatchTST, RMLP, Informer, Stationary, and Cross-former on PEMS (PEMS03, PEMS04, and PEMS08) datasets. We set the input length $I$ as 336 for all of these base models.

| | Models | Metric | PEMS03 | | | | PEMS04 | | | | PEMS08 | | | |
|---|---|---|---|---|---|---|---|---|---|---|---|---|---|---|
| | | | 96 | 192 | 336 | 720 | 96 | 192 | 336 | 720 | 96 | 192 | 336 | 720 |
| PatchTST | Original | MSE | 0.180 | 0.207 | 0.223 | 0.291 | 0.195 | 0.218 | 0.237 | 0.321 | 0.239 | 0.292 | 0.314 | 0.372 |
| | | MAE | 0.281 | 0.295 | 0.309 | 0.364 | 0.296 | 0.314 | 0.329 | 0.394 | 0.324 | 0.351 | 0.374 | 0.425 |
| | +TAM | MSE | 0.159 | 0.189 | 0.193 | 0.263 | 0.170 | 0.198 | 0.204 | 0.257 | 0.186 | 0.244 | 0.257 | 0.307 |
| | | MAE | 0.270 | 0.293 | 0.286 | 0.350 | 0.276 | 0.297 | 0.299 | 0.345 | 0.289 | 0.324 | 0.320 | 0.378 |
| | +InfoTime | MSE | **0.115** | **0.154** | **0.164** | **0.198** | **0.110** | **0.118** | **0.129** | **0.149** | **0.114** | **0.160** | **0.177** | **0.209** |
| | | MAE | **0.223** | **0.251** | **0.256** | **0.286** | **0.221** | **0.224** | **0.237** | **0.261** | **0.218** | **0.243** | **0.241** | **0.281** |
| RMLP | Original | MSE | 0.160 | 0.184 | 0.201 | 0.254 | 0.175 | 0.199 | 0.210 | 0.255 | 0.194 | 0.251 | 0.274 | 0.306 |
| | | MAE | 0.257 | 0.277 | 0.291 | 0.337 | 0.278 | 0.294 | 0.306 | 0.348 | 0.279 | 0.311 | 0.328 | 0.365 |
| | +TAM | MSE | 0.143 | 0.171 | 0.186 | 0.234 | 0.153 | 0.181 | 0.189 | 0.222 | 0.158 | 0.215 | 0.236 | 0.264 |
| | | MAE | 0.241 | 0.264 | 0.276 | 0.316 | 0.259 | 0.280 | 0.289 | 0.321 | 0.255 | 0.288 | 0.302 | 0.333 |
| | +InfoTime | MSE | **0.117** | **0.159** | **0.146** | **0.204** | **0.103** | **0.114** | **0.130** | **0.154** | **0.116** | **0.156** | **0.175** | **0.181** |
| | | MAE | **0.228** | **0.252** | **0.246** | **0.285** | **0.211** | **0.219** | **0.236** | **0.264** | **0.215** | **0.235** | **0.242** | **0.255** |
| Informer | Original | MSE | 0.139 | 0.152 | 0.165 | 0.216 | 0.132 | 0.146 | 0.147 | 0.145 | 0.156 | 0.175 | 0.187 | 0.264 |
| | | MAE | 0.240 | 0.252 | 0.260 | 0.290 | 0.238 | 0.249 | 0.247 | 0.245 | 0.262 | 0.266 | 0.274 | 0.325 |
| | +TAM | MSE | 0.126 | 0.142 | 0.157 | 0.207 | 0.118 | 0.128 | 0.134 | 0.138 | 0.126 | 0.149 | 0.172 | 0.237 |
| | | MAE | 0.230 | 0.245 | 0.254 | 0.284 | 0.226 | 0.232 | 0.235 | 0.240 | 0.232 | 0.247 | 0.265 | 0.316 |
| | +InfoTime | MSE | **0.109** | **0.120** | **0.144** | **0.194** | **0.107** | **0.124** | **0.124** | **0.136** | **0.099** | **0.123** | **0.147** | **0.196** |
| | | MAE | **0.216** | **0.228** | **0.247** | **0.282** | **0.215** | **0.230** | **0.231** | **0.245** | **0.204** | **0.224** | **0.242** | **0.278** |
| Stationary | Original | MSE | 0.120 | 0.143 | 0.156 | 0.220 | 0.109 | 0.116 | 0.129 | 0.139 | 0.151 | 0.180 | 0.252 | 0.223 |
| | | MAE | 0.222 | 0.242 | 0.252 | 0.300 | 0.214 | 0.220 | 0.230 | 0.240 | 0.235 | 0.247 | 0.262 | 0.285 |
| | +TAM | MSE | 0.118 | 0.143 | 0.156 | 0.208 | 0.104 | 0.115 | 0.123 | 0.136 | 0.134 | 0.160 | 0.191 | 0.231 |
| | | MAE | 0.219 | 0.241 | 0.252 | 0.285 | 0.209 | 0.218 | 0.223 | 0.234 | 0.224 | 0.237 | 0.251 | 0.289 |
| | +InfoTime | MSE | **0.101** | **0.131** | **0.153** | **0.190** | **0.096** | **0.114** | **0.125** | **0.135** | **0.103** | **0.144** | **0.184** | **0.217** |
| | | MAE | **0.206** | **0.229** | **0.245** | **0.273** | **0.199** | **0.217** | **0.229** | **0.243** | **0.200** | **0.220** | **0.245** | **0.278** |
| Crossformer | Original | MSE | 0.159 | 0.233 | 0.275 | 0.315 | 0.149 | 0.216 | 0.230 | 0.276 | 0.141 | 0.162 | 0.199 | 0.261 |
| | | MAE | 0.270 | 0.319 | 0.351 | 0.383 | 0.261 | 0.320 | 0.324 | 0.369 | 0.253 | 0.269 | 0.306 | 0.355 |
| | +TAM | MSE | 0.134 | 0.179 | 0.217 | 0.264 | 0.133 | 0.171 | 0.186 | 0.240 | 0.112 | 0.136 | 0.156 | 0.196 |
| | | MAE | 0.237 | 0.270 | 0.298 | 0.335 | 0.237 | 0.270 | 0.284 | 0.329 | 0.220 | 0.235 | 0.252 | 0.289 |
| | +InfoTime | MSE | **0.119** | **0.166** | **0.189** | **0.223** | **0.114** | **0.139** | **0.161** | **0.171** | **0.088** | **0.108** | **0.134** | **0.171** |
| | | MAE | **0.217** | **0.250** | **0.265** | **0.293** | **0.215** | **0.236** | **0.258** | **0.275** | **0.190** | **0.206** | **0.222** | **0.251** |

