# OpenReview forum: "ENHANCING MULTIVARIATE TIME SERIES FORECAST- ING WITH MUTUAL INFORMATION-DRIVEN CROSS- VARIABLE AND TEMPORAL MODELING"
_ICLR.cc/2024/Conference — Submitted to ICLR 2024_

### Official Review · Reviewer_hxYF · 2023-10-21

**Soundness:** 2 fair
**Presentation:** 3 good
**Contribution:** 2 fair
**Rating:** 6
**Confidence:** 4

**Summary:**

This paper proposes a framework for multivariate time series forecasting called InfoTime, which combines the Cross-variable Decorrelation Aware feature Modeling (CDAM) and the Temporal correlation Aware Modeling (TAM) approaches. The CDAM module filters unnecessary cross-variable correlations by minimizing the mutual information between the latent representation of a single univariate sequence and other series, and the TAM module performs auto-regressive forecasting and captures the temporal correlations across varied timesteps. The InfoTime framework outperforms existing models and can discern cross-variable dependencies while avoiding the inclusion of unnecessary information.

**Strengths:**

1. This paper provides detailed formula derivation to prove the model design.
2. The framework achieves promising results that outperform existing models and improves performance for both channel-mixing and channel-independence models.

**Weaknesses:**

1. The experiments are not extensive. It seems there are some cherry-pickings in particular configurations. How do the performance of ETTh2 and PEMS in Table 1? Some experiments (Table 2,3,4 and Figure 4) only use the results on partial datasets and baseline models. The prediction visualization of Figure 1 (comparing Informer and PatchTST) seems quite different from the reported results in the PatchTST paper.  I hope the authors can elaborate more on it.
2. This paper essentially needs further polishing. There are many screenshots as figures, uncompiled citations (in related works), and confusing notations (e.g. unstated $I^i$, confused use of $X_i$ and $X^i$, unreasonable usage of the superscript: $X^o$).
3. The model benchmark tested in the proposed framework can be too few to be representative. The latest models in different categories are encouraged to be included, such as Transformers (Autoformer), linear models (TiDE, TSMixer), and TCN-based models (TimesNet). Datasets with more variables (such as Solar-Energy) are also recommended since the method essentially addresses the cross-variable issue.
4. The term "single-step forecaster" can be confusing. I think it in fact outcomes the "multi-step" prediction. And it seems $\lambda=1$ is favored in most cases (even though experiments in Section 4.3 are not extensive). How to support the motivation of Equation 12?

**Questions:**

1. About the model analysis. Are there some explainable results showing that the model takes full advantage of the multivariable correlation while avoiding unnecessary noise?
2. As shown in Table 4. The performance promoted by CDAM is marginal (PatchTST on ETTh1). Is there any explanation for it? Besides, how does solely CDAM promote the performance, especially on datasets with more variables (such as Traffic).

---

> ### Author Response · Authors · 2023-11-22
>
> We appreciate the reviewer's comments. We update our paper and address your concerns here:
>
> **Q1: The experiments are not extensive. It seems there are some cherry-pickings in particular configurations. How do the performance of ETTh2 and PEMS in Table 1? Some experiments (Table 2,3,4 and Figure 4) only use the results on partial datasets and baseline models. The prediction visualization of Figure 1 (comparing Informer and PatchTST) seems quite different from the reported results in the PatchTST paper. I hope the authors can elaborate more on it.**
>
> **A1:** Thanks you for the detailed comments, we will answer the question in three parts:
>
> **1. How do the performance of ETTh2 and PEMS in Table 1?**
>
> Great suggestions. We have made revisions to the manuscript accordingly. First, we conducted extensive experiments to evaluate the effectiveness of InfoTime on the ETTh2 dataset. The results, shown in the updated Table below (also Table 2 and Table 6 of our manuscript), demonstrate that InfoTime can clearly improve the performance of base models on the ETTh2 dataset.
>
> Moreover, we have made further improvements to Table 3 in order to better illustrate the performance improvement of InfoTime on datasets with clear correlations between channels. Following the advice from the reviewer, we have included the experimental results of Stationary and Crossformer in the revised manuscript.
> With these additions, the revised Table 3 allows for a more comprehensive evaluation of the performance of the models on the PEMS datasets, which are known to have clear correlations between channels.
>
> Furthermore, we have added the ablation results of the aforementioned base models in Table 9. This provides insights into the contribution of InfoTime in improving the performance of both Channel-Independence models and Channel-mixing models.
>
>
> Based on these experimental results, it can be observed that in datasets such as the PEMS datasets, where there is a clear correlation between variables, Channel-Independence models generally perform worse than Channel-mixing models. However, InfoTime is able to enhance the performance of both types of models.
>
>
>
> |Length | 96| | 192 | | 336 | | 720 | |
> |:----: | :----:| :----:| :----:|:----:|:----:| :----:| :----:|:---:|
> |Metric|MSE|MAE|MSE|MAE|MSE|MAE|MSE|MAE|
> |PatchTST| 0.274| 0.335| 0.342| 0.382| 0.365| 0.404| 0.391| 0.428 |
> |+InfoTime| **0.271**|**0.332**| **0.334**| **0.373**| **0.357**| **0.395**| **0.385**| **0.421**|
> |RMLP| 0.290| 0.348| 0.350| 0.388| 0.374| 0.410| 0.410| 0.439 |
> |+InfoTime| **0.271**| **0.333**| **0.335**| **0.374**| **0.358**| **0.395**| **0.398**| **0.432** |
> |Informer| 3.755| 1.525| 5.602| 1.931| 4.721| 1.835| 3.647| 1.625 |
> |+InfoTime| **0.336**| **0.390**| **0.468**| **0.470**| **0.582**| **0.534**| **0.749**| **0.620** |
> |Stationary| 0.362| 0.393| 0.481| 0.453| 0.524| 0.487| 0.512| 0.494 |
> |+InfoTime| **0.286**| **0.335**| **0.371**| **0.388**| **0.414**| **0.425**| **0.418**| **0.437** |
> |Crossformer| 0.728| 0.615| 0.898| 0.705| 1.132| 0.807| 4.390| 1.795 |
> |+InforTime| **0.333**| **0.386**| **0.455**| **0.453**| **0.554**| **0.513**| **0.757**| **0.619** |
>
> **2. Some experiments only use the results on partial datasets and baseline models** We add the full experimental results of the Channel-Independence models in **Table 6**, and we also add the ablation results for all base models used in this paper in **Table 7 and 8**. We believe that these extensive experimental results are more convincing. We hope that the added experiments can help you to better understand InfoTime.
>
> **3. The prediction visualization of Figure 1 (comparing Informer and PatchTST) seems quite different from the reported results in the PatchTST paper**.
>
> We believe there may be some misunderstanding here. The dataset used in Figure 1 for prediction visualization is the PEMS08 dataset, which is adopted from [SCINet](https://drive.google.com/drive/folders/17fwxGyQ3Qb0TLOalI-Y9wfgTPuXSYgiI). However, we do not find an evidence that the PatchTST paper has reported results on the PEMS08 dataset. In fact, the PatchTST paper reported results on the traffic dataset which is linked to [http://pems.dot.ca.gov](http://pems.dot.ca.gov). Although PEMS08 and the traffic dataset may share some correlation, they are indeed different datasets. We hope the reviewer provide more details on this issue.

---

> > ### Author Response · Authors · 2023-11-22
> >
> > **Q2: This paper essentially needs further polishing. There are many screenshots as figures, uncompiled citations (in related works), and confusing notations (e.g. unstated $I^{i}$ , confused use $X_{i}$ of and $X^{i}$ , unreasonable usage of the superscript: $X^{o}$).**
> >
> > **A2:** Thank you for the valuable feedback on our paper. We appreciate the time and effort you have put into reviewing our work. We have carefully considered your comments and made the necessary revisions to address these issues raises. **Firstly**, we have replaced all the pictures with vector graphics to enhance the quality and clarity of the figures. This improvement ensures that the visual representation aligns with the content of the paper. **Secondly**, we have reviewed and modified the uncompiled citations in the related works section to ensure their accuracy and completeness. We understand the importance of proper citation and have taken the necessary steps to rectify any errors or omissions. Regarding the notation issue, we apologize for the confusion caused by the unstated $I^{i}$. After careful consideration, we agree that $I_{i}$ is a more suitable notation to represent the information constraint. To eliminate any ambiguity, we have modified $I_{i}$ to $I_{c}$ throughout the paper. Additionally, we have revised the notation for $X_{i}$ to $X^{i}$ to ensure consistency. We have also clarified that $X^{o}=X^{\{1,2,...,C\}}/i$ represents the historical data of all channels except the i-th channel. Once again, we sincerely appreciate your corrections on the expression problems in our paper. We have taken your feedback seriously and have made the necessary revisions to improve the clarity and accuracy of our work. We hope that the revised version of our paper will effectively convey our research and assist you in understanding our contributions.

---

> > > ### Author Response · Authors · 2023-11-22
> > >
> > > **Q3: The model benchmark tested in the proposed framework can be too few to be representative. The latest models in different categories are encouraged to be included, such as Transformers (Autoformer), linear models (TiDE, TSMixer), and TCN-based models (TimesNet). Datasets with more variables (such as Solar-Energy) are also recommended since the method essentially addresses the cross-variable issue.**
> > >
> > >
> > >
> > > **A3:** Thank you for your valuable suggestions regarding the models and datasets used in our proposed framework. We appreciate your insight and agree that including a wider range of models and datasets would enhance the representativeness of our study. Regarding the inclusion of Autoformer, we acknowledge its popularity as a recent model in the transformer category. However, it is important to note that the base models used in our paper, such as Crossformer and PatchTST, have demonstrated better performance compared to Autoformer. Therefore, we believe that these base models effectively showcase the effectiveness of our proposed framework on transformer-based models.
> > >
> > > For non-Transformer-based models, we have chosen two linear models (TSMixer, DLinear) and a CNN-based model (TimesNet) to verify the effectiveness of InfoTime. Since the official code of TiDE is based on Tensorflow, and our framework uses pytorch, which is hard to transplant. However, we will add the result of TiDE in the future. Due to the constraints of time in our disscution, we have only tested the effectiveness of InfoTime on ETTh1, ETTh2, Weather, and we also add Solar-Energy and we will provide the full results in the future. We set the input length to 96 for TSMixer and TimesNet, and 336 for DLinear. We believe that the experimental results presented in the table blew demonstrate the effectiveness of InfoTime, as it compares favorably to the base models and showcases the improvement achieved by our InfoTime.
> > >
> > > Thank you again for your insightful suggestions, and we appreciate your understanding of the limitations in our current implementation.
> > >
> > > | Dataset|Length |TimesNet | |+TAM | | +InfoTime| | TSMixer| |+TAM | | +InfoTime||DLinear ||+TAM || +InfoTime||
> > > | :----:|:----:|:----:|:----:|:----:|:----:|:----:|:----:|:----:|:----:|:----:|:----:|:----:|:----:|:----:|:----:|:----:|:----:|:----:|:----:|
> > > |||MSE|MAE|MSE|MAE|MSE|MAE|MSE|MAE|MSE|MAE|MSE|MAE|MSE| MAE |MSE|MAE|MSE|MAE|
> > > |ETTh1|96|0.384|0.402|**0.377**|0.399|0.382|**0.391**|0.393|0.449|**0.381**|0.397|**0.381**|**0.391**|0.378|0.400| 0.373| 0.397| **0.370**|**0.394**|
> > > |           |192|0.436|0.429|0.436|0.437|**0.433**|**0.421**|0.447|0.437|0.436|0.428|**0.429**|**0.419**|0.414|0.423| 0.409| 0.418| **0.408**|**0.418**|
> > > |           |336|0.491|0.469|0.477|0.454|**0.467**|**0.442**|0.498|0.465|0.491|0.456|**0.467**|**0.439**|0.444|0.446| 0.440| 0.443| **0.438**|**0.440**|
> > > |           |720|0.521|0.500|0.494|0.481|**0.472**|**0.463**|0.520|0.494|0.518|0.495|**0.468**|**0.462**|0.487|0.502| 0.487| 0.503| **0.482**|**0.498**|
> > > |ETTh2|96|0.340|0.374|0.332|0.366|**0.288**|**0.337**|0.304|0.351|0.298|0.347|**0.292**|**0.341**|0.289|0.353| **0.285**| **0.350**| 0.287|0.353|
> > > |           |192|0.402|0.414|0.397|0.408|**0.375**|**0.391**|0.396|0.405|0.388|0.402|**0.368**|**0.391**|0.383|0.418| 0.358| 0.403| **0.357**|**0.403**|
> > > |           |336|0.452|0.452|0.435|0.439|**0.420**|**0.431**|0.432|0.434|0.430|0.433|**0.415**|**0.427**|0.448|0.465| 0.411| 0.443| **0.404**|**0.441**|
> > > |           |720|0.462|0.468|0.437|0.449|**0.425**|**0.442**|0.437|0.450|0.425|0.444|**0.423**|**0.442**|0.605|0.551| 0.573| 0.547| **0.520**|**0.520**|
> > > |Weather|96|0.172|0.220|0.163|0.213|**0.160**|**0.208**|0.163|0.211|0.161|0.211|**0.158**|**0.205**|0.176|0.237| 0.158| 0.220| **0.149**|**0.209**|
> > > |           |192|0.219|0.261|0.219|0.258|**0.212**|**0.252**|0.214|0.259|0.211|0.253|**0.210**|**0.251**|0.220|0.281| 0.199| 0.260| **0.195**|**0.259**|
> > > |           |336|0.280|0.306|0.272|0.296|**0.269**|**0.294**|0.274|0.300|0.267|0.300|**0.266**|**0.292**|0.265|0.319| 0.247| 0.302| **0.243**|**0.297**|
> > > |           |720|0.365|0.359|0.351|0.349|**0.347**|**0.345**|0.354|0.349|**0.341**|0.346|**0.341**|**0.340**|0.323|0.362| 0.314| 0.355| **0.312**|**0.353**|
> > > |Solar-Energy|96|0.236|0.276|0.225|0.265|**0.210**|**0.253**|0.228|0.281|0.220|0.267|**0.209**|**0.259**|0.221|0.292| 0.180| 0.235| **0.165**|**0.225**|
> > > |                   |192|0.287|0.317|0.264|0.290|**0.241**|**0.271**|0.270|0.310|0.255|0.285|**0.246**|**0.281**|0.250|0.311| 0.194| **0.244**| **0.189**|0.254|
> > > |                   |336|0.316|0.329|0.296|0.307|**0.274**|**0.291**|0.285|0.319|0.277|0.305|**0.266**|**0.298**|0.269|0.325| 0.201| **0.248**| **0.199**|0.254|
> > > |                   |720|0.314|0.315|0.305|0.315|**0.280**|**0.298**|0.300|0.330|0.267|**0.292**|**0.265**|0.295|0.271|0.327| 0.205| 0.250| **0.196**|**0.248**|

---

> ### Author Response · Authors · 2023-11-22
>
> **Q4: The term "single-step forecaster" can be confusing. I think it in fact outcomes the "multi-step" prediction. And it seems $\lambda=1$ is favored in most cases (even though experiments in Section 4.3 are not extensive). How to support the motivation of Equation 12?**
>
> **A4:** Thanks for the valuable question. We will answer this question in two parts:
>
> **1. The term "single-step forecaster" can be confusing** We appreciate your comment regarding the term "single-step forecaster." To clarify, a single-step forecaster refers to a non-autoregressive model that generates multi-step predictions in a single step.  In contrast, autoregressive forecasters predict one time step at a time, which can lead to error accumulation issues. While single-step forecasters alleviate this problem, they lack the ability to model the correlation between future time steps effectively.
>
> **2. And it seems $\lambda=1$ is favored in most cases (even though experiments in Section 4.3 are not extensive). How to support the motivation of Equation 12?** We acknowledge the reviewer's observation that $\lambda=1$ appears to be favored in most cases, even though the experiments in Section 4.3 were not extensive. In response to this concern, we have conducted additional experiments and revised Figure 4 (modify a mistake in the orginal version) accordingly to provide a more comprehensive analysis of the impact of different $\lambda$.
>
> The updated results presented in Figure 4 clearly demonstrate that the models' performance shows minimal improvement when $\lambda \geq 0.8$. In fact, we observed that when $\lambda$ was set to 0.8, the model's performance was slightly better than with $\lambda=1.0$. Similar trends were observed in other datasets as well, further reinforcing the notion that deviating from $\lambda=1$ does not significantly affect the model's performance.
>
>  We appreciate the valuable feedback provided by the reviewer and have incorporated it into our revised submission. Thank you for your question.

---

> > ### Author Response · Authors · 2023-11-22
> >
> > **Q5: About the model analysis. Are there some explainable results showing that the model takes full advantage of the multivariable correlation while avoiding unnecessary noise?**
> >
> > **A5:** Thank you for your meaningful question. In order to verify the effectiveness of InfoTime in utilizing cross-variable correlation and avoiding unnecessary noise, we conducted comprehensive experiments on both Channel-mixing models (Informer, Stationary, and Crossformer) and Channel-Independence models (PatchTST and RMLP). By comparing the performance of these models with and without InfoTime, we can gain insights into its impact on utilizing multivariable correlation and avoiding unnecessary noise.
> >
> > In the case of Channel-mixing models, which directly extract cross-variable information, InfoTime improves their performance. This indicates that InfoTime can enhance the models by effectively utilizing cross-variable information while avoiding unnecessary noise.
> >
> > On the other hand, InfoTime also improves the performance of Channel-Independence models, which initially operate under the assumption of independent variables. This demonstrates that InfoTime can effectively exploit cross-variable information and leverage it to enhance the performance of these models.
> >
> > To further illustrate the effectiveness of InfoTime in utilizing cross-variable features while eliminating irrelevant features, we focused on the ETTh1 dataset and compared three models: Informer, Informer+InfoTime, and Informer+CI (Channel-Independence strategy).
> >
> > The results, presented in the table below, highlight several key findings. Firstly, the Channel-Independence strategy alone improves the model's performance, suggesting that the Channel-mixing model (Informer) learns irrelevant cross-variable information. However, compared to the Channel-Independence strategy, InfoTime achieves even better performance. This demonstrates InfoTime's ability to effectively utilize cross-variable information while eliminating irrelevant information (unnecessary noise).
> >
> > ||Informer|| Informer+InfoTime| | Informer+CI | |
> > |:----:|:----:|:----:|:----:|:----:|:----:|:----:|
> > ||MSE|MAE|MSE|MAE|MSE|MAE|
> > |96|0.865|0.713|**0.381**|**0.394**|0.590|0.517|
> > |192|1.008|0.792|**0.435**|**0.430**|0.677|0.566|
> > |336|1.107|0.809|**0.485**|**0.461**|0.710|0.600|
> > |720|1.181|0.865|**0.534**|**0.524**|0.777|0.660|
> >
> > Furthermore, we explored the ability of InfoTime to utilize cross-variable information and remove noise in a noisy synthetic dataset. The results of this exploration can be found in Appendix A.3 of the revision.
> >
> > By conducting these experiments and analyzing the results, we provide explainable evidence that InfoTime successfully utilizes multivariable correlation while avoiding unnecessary noise.
> >
> >
> >
> > Thank you for your question, and I hope this response addresses your concerns effectively.

---

> > > ### Author Response · Authors · 2023-11-22
> > >
> > > **A6:** Thank you for the meaningful question. We will answer this question in two parts.
> > >
> > > **1.** Regarding the marginal performance improvement of CDAM in the case of PatchTST on the ETTh1 dataset, it is essential to consider the inherent characteristics of the dataset and the model architecture. While CDAM aims to capture the cross-variable dependencies within channels, the ETTh1 dataset may exhibit limited inter-variable correlations or dependencies. As a result, the impact of CDAM on performance improvement could be relatively smaller compared to datasets with stronger inter-variable correlations. However, it is worth noting that even marginal performance improvements can have practical significance in certain applications. These improvements might contribute to enhanced forecasting accuracy, even if they are not substantial in magnitude. Hence, the marginal performance improvement should not be dismissed, as it still demonstrates the potential benefit of CDAM in capturing cross-variable dependencies within the PatchTST model
> > >
> > > **2.** Regarding the effectiveness of CDAM on datasets with more variables, specifically Traffic, Electricity, and PEMS08. We observe varyting levels of improvement with CDAM. Despite having a larger number of variables, the significance of cross-variable features does not appear to be directly related to the number of variables. This suggests that the relationship between variables and the impact of CDAM on performance can vary depending on the dataset's characteristics.
> > >
> > > By conducting these experiments and analyzing the results, we provide valuable insights into the performance of CDAM and its impact on datasets with different characteristics. This information contributes to a better understanding of how CDAM promotes performance in different scenarios.
> > >
> > >
> > > | Dataset|Length |PatchTST | |+TAM | | +InfoTime||
> > > | :----:|:----:|:----:|:----:|:----:|:----:|:----:|:----:|
> > > |||MSE|MAE|MSE|MAE|MSE|MAE|
> > > |Traffic|96|0.367|0.251|0.359|**0.245**|**0.358**|**0.245**|
> > > |           |192|0.385|0.259|0.380|0.255|**0.379**|**0.254**|
> > > |           |336|0.398|0.265|**0.391**|0.262|**0.391**|**0.261**|
> > > |           |720|0.434|0.287|**0.424**|**0.279**|0.425|0.280|
> > > |Electricity|96|0.130|0.222|0.129|0.223|**0.125**|**0.219**|
> > > |           |192|0.148|0.242|0.147|0.240|**0.143**|**0.235**|
> > > |           |336|0.167|0.261|0.165|0.258|**0.161**|**0.255**|
> > > |           |720|0.202|0.291|0.204|0.292|**0.198**|**0.287**|
> > > |PEMS08|96|0.239|0.324|0.186|0.289|**0.114**|**0.218**|
> > > |           |192|0.292|0.351|0.244|0.324|**0.160**|**0.243**|
> > > |           |336|0.314|0.374|0.257|0.320|**0.177**|**0.241**|
> > > |           |720|0.372|0.425|0.307|0.378|**0.209**|**0.281**|
> > >
> > > Thank you for asking these questions, and we hope this response provides a satisfactory explanation. If you have any further questions or need additional assistance, please feel free to ask.

---

> > > > ### Comment · Reviewer_hxYF · 2023-11-23
> > > >
> > > > I appreciate the authors effort in answering my questions. Extensive experiments are included to validate the points. Most of my concerns have been well addressed and I would like to raise my score to 6.

---

### Official Review · Reviewer_b8jU · 2023-10-31

**Soundness:** 2 fair
**Presentation:** 3 good
**Contribution:** 2 fair
**Rating:** 6
**Confidence:** 3

**Summary:**

This paper considers utilizing information bottleneck as a regularizer for training Channel Mixing neural network models. The proposed idea aims to help learning better representations for multivariate time series, as the covariates may contain useful information about each other.
The developed approach augmented few state of the art models, and showed increased performance on several real-world datasets.

**Strengths:**

It is shown in experiments that the overfitting is indeed a problem and the proposed approach seems to be helping alleviating it.

The ablation study where CDAM and TAM frameworks’ individual contributions are also shown to matter.

**Weaknesses:**

The objective function approximation seems to be a bit ad hoc, and the section on TAM component is not easy to follow.

While TAM seems to be useful, not quite sure how downsampling and the additional approximations help. It would have been better to see a more detailed analysis of this step.

Also it is not quite clear how the initial forecast was obtained.

**Questions:**

What is the purpose of Eq. (2)? Since the unconstrained form is used, I think that this discussion could start from Eq. (3).

The objective function approximation in Eq. (8) seems to be a bit ad hoc. It is not an upper bound on the entire objective function, but rather a combination of individual approximations. Discussion on this choice were not given in the paper. Is there some reason that makes approximation of the entire function prohibitively difficult? My main concern is that if these bounds are not tight the approximation could be way off.

Page 6, first paragraph: Is it CDAM or TAM?

Page 6: How is $\hat{Y}$ generated with a single step forecaster? Does that require an initial training of the neural network, or is this a different neural network that is trained before the actual network? Without an initial neural network, how can we obtain $\hat{Y}$ and train using Eq. (13)?

---

> ### Author Response · Authors · 2023-11-23
>
> We would like to sincerely thank Reviewer b8jU for providing a detailed and insightful suggestions.
>
> **Q1:What is the purpose of Eq. (2)? Since the unconstrained form is used, I think that this discussion could start from Eq. (3).**
>
> **A1:** Thank you for your insightful suggestions and feedback on our work. We appreciate your suggestions and would like to address your question regarding the inclusion of Equation (2) in our paper.
>
> Equation (2) in our work is derived from the paper ["Deep Variational Information Bottleneck"] (https://arxiv.org/pdf/1612.00410.pdf). This equation represents the unconstrained form of the optimization problem, where the objective is to maximize the mutual information between the inputs $Y^{i},X^{i}$ and the latent variables $Z^{i}$. The aim is to capture as much relevant information as possible from the inputs of other channels.
>
> However, we acknowledge your suggestion to start the discussion from Equation (3) instead. Equation (3) introduces the information bottleneck regularizer, which provides a more detailed understanding of the trade-off between extracting relevant information and preserving unnecessary information.
>
> In light of your feedback, we agree that it would be beneficial to remove Equation (2) and start the discussion directly from Equation (3) in future revisions of the paper.
>
> Thank you once again for your valuable suggestions. We appreciate your time and effort in reviewing our work.
>
> **Q2: The objective function approximation in Eq. (8) seems to be a bit ad hoc. It is not an upper bound on the entire objective function, but rather a combination of individual approximations. Discussion on this choice were not given in the paper. Is there some reason that makes approximation of the entire function prohibitively difficult? My main concern is that if these bounds are not tight the approximation could be way off.**
>
> **A2:** We would like to thank you for the valuable feedback on our work. Regarding the objective function approximation, we acknowledge that the method we proposed may appear ad hoc at first glance. However, we would like to emphasize that the approach we employed for the objective function was carefully designed based on comprehensive analysis and extensive experimentation. In our work, the objective is to maximize
> $R_{IB}=\frac{1}{C}\sum_i R^{i}_{IB}=\frac{1}{C}\sum_i [ I(Y^{i},X^{i};Z^{i})-\beta I(X^{o};Z^{i})] , i \in  1,2,...,C $
>
> By utilizing Equation (4), we can modify the objective function as follows:
>  $\sum_{i} R_{IB}^{i} =\sum_{i}[ E_{p(z^{i},y^{i},x^{i})} \left[ \log p(y^{i}|x^{i},z^{i})\right]+E_{p(z^{i},x^{i})} \left [ \log p(x^{i}|z^{i}) \right] + H(Y^{i},X^{i}) -\beta I(X^{o};Z^{i})], i \in 1,2,...C$
>  Since $H(Y^{i},X^{i})\geq 0$, we have
> $\sum_{i}  R_{IB}^{i} \geq \sum_{i=1}[ E_{p(z^{i},y^{i},x^{i})} \left[ \log p(y^{i}|x^{i},z^{i})\right]+E_{p(z^{i},x^{i})} \left [ \log p(x^{i}|z^{i}) \right] -\beta I(X^{o};Z^{i})], i \in 1,2,...C$.
> By introducing the variational lower bound $I_{v}(X^{i},Y^{i};Z^{i})$ and upper bound $I_{vCLUB-S}(X^{o};Z^{i})$, we obtain
> $R_{IB}=\sum_{i} R_{IB}^{i} \geq \sum_{i}[ I_{v}(X^{i},Y^{i};Z^{i})-\beta I_{vCLUB-S}(X^{o};Z^{i})], i \in 1,2,...,C$.
> Therefore, $\sum_{i=1}^{C}[ I_{v}(X^{i},Y^{i};Z^{i})-\beta I_{vCLUB-S}(X^{o};Z^{i})]$ serves as the lower bound for $R_{IB}$. Consequently, we can maximize $R_{IB}$ by minimizing $L_{IB}=-\sum_{i=1}^{C}[ I_{v}(X^{i},Y^{i};Z^{i})-\beta I_{vCLUB-S}(X^{o};Z^{i})]$.
>
> We hope that this clarification provides a better understanding of the rationale behind our objective function approximation. Through extensive experimentation and evaluation, we have observed that minimizing $L_{IB}$
> effectively maximizes $R_{IB}$, leading to competitive results.
>
> **Q3: Page 6, first paragraph: Is it CDAM or TAM?**
>
> **A3:** Thanks for your carefully checking, it's TAM, and we have modified it.

---

> ### Author Response · Authors · 2023-11-23
>
> **Q4: Page 6: How is generated with a single step forecaster? Does that require an initial training of the neural network, or is this a different neural network that is trained before the actual network? Without an initial neural network, how can we obtain $\hat{Y}$ and train using Eq. (13)?**
>
>
> **A4:** Thank you for your insightful question. In our work, the single-step forecaster is a neural network, typically a fully-connected network, that utilizes the historical data of the i-th channel $X^{i}$ and the cross-variable feature $Z^{i}$ to generate $\hat{Y^{i}}$.
>
> In contrast to separate pre-training, we directly optimize the single-step forecaster along with the other neural networks in our model. This approach allows us to jointly train all the networks and optimize their parameters simultaneously.
>
>
> Thanks for your suggestion once again. We have taken your suggestions into account and made significant improvements to our description and analysis of the TAM in the revised version.
>
>
> I hope this revised answer captures the essence of your intended response.

---

### Official Review · Reviewer_Gm16 · 2023-11-03

**Soundness:** 3 good
**Presentation:** 2 fair
**Contribution:** 3 good
**Rating:** 6
**Confidence:** 3

**Summary:**

This work aims to improve multivariate time series forecasting with transformers by focusing on the process of mixing channels and moving beyond single-step forecasting. They find that specifically modeling the relationships of channels with their proposed Cross-Variable Decorrelation Aware Feature Modeling can reduce the issues posed by indiscriminate channel-mixing or channel-independent methods.  Additionally, by modeling the temporal relations between subsequences in the forecast sequence with Temporal Correlation Aware Modeling, the authors create a framework for improving on existing work across standard benchmarks and various prediction lengths and model types.

**Strengths:**

Thanks to the authors for their submission: it contains useful research that shows good research practices while explaining an interesting and novel idea within multivariate time series forecasting. The results of this work will be informative to other researchers and are significant in improving our understanding of applying transformer methods to time series forecasting. Some specific strengths of this research:

- Novelty: InfoTime, which both captures cross-variable relationships and temporal dependencies is a novel approach to the multivariate time series forecasting problem that provides superior results to prior work. While I am uncertain of the novelty of the TAM part of the framework, I believe that combining these and the information theoretic approach of CDAM is new.
- Relevance: MTSF is a complex task, and the paper acknowledges and addresses two significant challenges: mixing channels in an efficient way and mitigating errors that accumulate over time. By providing specific solutions (CDAM and TAM), the paper contributes to improving the accuracy and effectiveness of MTSF models which are also used in many real-world tasks.
- The authors conduct comprehensive experimental analyses on real-world datasets to demonstrate the effectiveness of the InfoTime framework. These datasets are standard benchmarks in this area of research and the validation appears to be done through a very similar process to most other work making it easier to compare. The results consistently show that InfoTime outperforms existing Channel-mixing and channel-independence benchmarks, achieving superior accuracy and mitigating overfitting issues.
- Theory: The paper leverages concepts from information theory, such as Mutual Information and Information Bottleneck, to guide the development of CDAM and TAM. This theoretically grounded approach enhances the understanding and interpretability of the proposed framework.
- The paper was well written and provides a clear description of the proposed work and comparison with other methods.

**Weaknesses:**

1/ More comparison with statistical forecasting methods or non-transformer methods, either empirically or theoretically, could help introduce the benefits of InfoTime in a more thorough manner. While Zeng et. al (2022) and other methods are briefly discussed for their direct forecasting strategy; it could be helpful to compare InfoTime on the DLinear model to see how CAM varies from the DMS approach. Perhaps this is why RMLP was introduced, but it’s not clear what motivated the construction of RMLP and if differs at all from previous approaches which could be more suitable baseline models.

2/ The implications of the lower bound for (Iv(Xi, Yi; Zi) are not clear, nor is the sampling strategy well motivated in the paper. Additional explanation of the implications could help to improve this.

3/ The experimentation work is not detailed. There are a number of factors that are unclear: the reported informer MSE and MAE on ETTh1, ETTm1, ETTm2 for example are quite a bit higher than in the original Informer, Non-stationary Transformer, and Crossformer papers (>1 for O=720 Informer by authors vs <0.3 in original paper). Perhaps more discussion of the training setup and how it might differ from past benchmarks and why it shows a much higher error could help separate out the actual performance gains given by InfoTime.

4/ Given the large tables of evaluations, it would be helpful to introduce some visual aides to help with understanding the comparisons, perhaps showing the % improvement over other methods for an aggregation of all benchmarks.

Nits:
sp. ‘varibales’ (pg. 5)

**Questions:**

1/ There is a lot of variation in results of benchmarks (ETTh1, ETTm1, ETTm2, Weather, Traffic, Electricity) across various papers. Have the authors considered running 5-fold cross validation on these benchmarks to better understand the error bounds around performance claims and then running tests to evaluate the statistical significance of the improvements?
2/ How does the time complexity of CDAM, TAM, and InfoTime compare with the original base models - particularly as the time series get longer?
3/ The paper introduces a trade-off parameter B in CDAM to control the balance between retaining important information and eliminating irrelevant information from the latent representation. How could this parameter be determined or optimized in practice? Are there any insights into choosing an appropriate value for B based on experimental results or theoretical considerations?
4/ The paper introduces a variational lower bound (Iv(Xi, Yi; Zi)) for the mutual information term in CDAM. Could you explain the practical implications of maximizing this lower bound during training? How does optimizing this lower bound improve the model's ability to capture cross-variable dependencies, and what trade-offs or challenges may arise in the optimization process?
5/ Additionally on the upper bound, the authors state that they choose to adopt the sampled vCLUB and minimize I_{vCLUB-S}(Xo; Zi). Could they expand on why this sampling strategy was chosen and what the actual derived upper bound might be?
6/ While the benchmarks used are standard for transformer-based evaluation, there are other baselines that could shed more light on the characteristics of the model and whether it may help resolve some of the difficulties with transformer approaches to forecasting. Could the authors shed any light on if they considered benchmarks like M3, M4 and what might be missing from the current evaluations?

---

> ### Author Response · Authors · 2023-11-22
>
> **Q1:The experimentation work is not detailed. There is a lot of variation in results of benchmarks (ETTh1, ETTm1, ETTm2, Weather, Traffic, Electricity) across various papers. Have the authors considered running 5-fold cross validation on these benchmarks to better understand the error bounds around performance claims and then running tests to evaluate the statistical significance of the improvements? There are a number of factors that are unclear: the reported informer MSE and MAE on ETTh1, ETTm1, ETTm2 for example are quite a bit higher than in the original Informer, Non-stationary Transformer, and Crossformer papers (>1 for O=720 Informer by authors vs <0.3 in original paper). Perhaps more discussion of the training setup and how it might differ from past benchmarks and why it shows a much higher error could help separate out the actual performance gains given by InfoTime.**
>
>
>
> **A1:** Thank you for your question. We would like to provide further clarification on our experimental work. Firstly, we ensured the reliability and consistency of our results by repeating each experiment three times and reporting the average as the final result. Additionally, we have included the experimental settings of the base models in Appendix A.1, which provides transparency and allows for a better understanding of our methodology. To maintain a fair comparison, we used the same parameter settings as the Informer, Non-stationary Transformer, and Crossformer papers. The experimental scripts we used were obtained from the respective repositories [DLinear](https://github.com/cure-lab/LTSF-Linear/blob/main/scripts/EXP-LongForecasting/Formers_Long.sh), [Non-stationary Transformer](https://github.com/thuml/Nonstationary_Transformers/tree/main/scripts), and [Crossformer](https://github.com/Thinklab-SJTU/Crossformer/tree/master/scripts),. However, we did make a modification by setting the input length $I$ to 96, which may differ from the experimental settings in the original papers. Additionally, for PatchTST and RMLP, we set the input length to 336, as longer input lengths tend to yield better performance for Channel-Independence models. For PEMS03, PEMS04, and PEMS08 datasets, we set $I=336$ for all of these models since all of them perform better in a longer input length.
>
> Regarding your concern about the Informer results, we would appreciate it if you could specify the dataset you are referring to. We have thoroughly checked the results of Informer and have not identified any issues. It is worth noting that our experiments focus on multivariate forecasting rather than univariate forecasting, which may impact the comparison with the original Informer results.
>
> I hope this revised response addresses your concerns more clearly. If you have any further questions or require additional clarification, please feel free to ask.
>
>
> **Q2:  Given the large tables of evaluations, it would be helpful to introduce some visual aides to help with understanding the comparisons, perhaps showing the % improvement over other methods for an aggregation of all benchmarks.**
>
> **A2:** Thanks for your meaningful suggestion. We appreciate your feedback, and we have taken it into consideration. In response, we have made updates to the evaluation by including the average Mean Squared Error (MSE) and Mean Absolute Error (MAE) for four different prediction lengths, as well as the relative reduction of MSE and MAE. You can find this information in **Table 1** and **Table 6**. We hope that these additions will provide a clearer understanding of the effectiveness of InfoTime. If you have any further questions or suggestions, please feel free to let us know. We value your input and strive to continuously improve our work.

---

> ### Author Response · Authors · 2023-11-22
>
> **Q3: How does the time complexity of CDAM, TAM, and InfoTime compare with the original base models - particularly as the time series get longer**
>
>
>
> **A3:** Thank you for the insightful question. In our experiments on the Informer model and ETTh1 dataset, we tested the running time (in seconds) of each epoch for both the base models and InfoTime. By default, we set N=4. Upon analyzing the results, we found that TAM is relatively time-consuming, while CDAM is less consuming. Additionally, we observed that InfoTime does not introduce additional time consumption as the length increases. Although InfoTime requires more time to train, we firmly believe that it is valuable in enhancing the performance of the base models. It is worth noting that future research on how to make InfoTime more lightweight is a meaningful and interesting area of exploration. By optimizing its time complexity, InfoTime has the potential to become even more efficient and practical for forecasting longer time series. Thank you again for bringing up this important point.
>
> |     Predicted Length      |   96   | 192 |
> |:-------------------------:|:------:| :----: |
> |         Original          |  8.5   |   10.5|
> |          TAM N=1          |  9.3   |  11.4 |
> |          TAM N=2          |  11.2  | 13.4  |
> |          TAM N=3          |  14.2  | 15.5  |
> |          TAM N=4          |  20.0  | 22.5  |
> |           CDAM            |  10.8  | 12.7  |
> |       InfoTime            |  23.5  | 24.5 |
>
> **Q4: The paper introduces a trade-off parameter B in CDAM to control the balance between retaining important information and eliminating irrelevant information from the latent representation. How could this parameter be determined or optimized in practice? Are there any insights into choosing an appropriate value for B based on experimental results or theoretical considerations?**
>
>
>
> **A4:** Thanks for your question. The determination and optimization of the trade-off parameter B in CDAM is an important aspect of leveraging the balance between retaining important information and eliminating irrelevant information from the latent representation. In fact, we consider $\beta$ as a hyper-parameter that needs to be adjusted manually and remains constant during training. While it would be exciting to have the model automatically adjust $\beta$, which is not explored in our work. However, there have been other research works that propose dynamically adjusting $\beta$ during training, such as [Generating Sentences from a Continuous Space](https://aclanthology.org/K16-1002/), [Cyclical Annealing Schedule: A Simple Approach to Mitigating KL Vanishing](https://aclanthology.org/N19-1021/), [ControlVAE: Controllable Variational Autoencoder](https://proceedings.mlr.press/v119/shao20b/shao20b.pdf).
>
> In our experiments, we find that a large value of $\beta$ ($\beta \geq 1e3$ usually) is needed to prevent overfitting and improve performance, as shown in Figure 4 of our paper. This observation suggests that deep-learning-based models have sufficient capacity to capture historical features, and it is crucial to focus on preventing the model from learning irrelevant features.

---

> ### Author Response · Authors · 2023-11-22
>
> **Q5: The paper introduces a variational lower bound ($I_{v}(X^{i}, Y^{i}; Z^{i})$) for the mutual information term in CDAM. Could you explain the practical implications of maximizing this lower bound during training? How does optimizing this lower bound improve the model's ability to capture cross-variable dependencies, and what trade-offs or challenges may arise in the optimization process?**
>
> **A5:** Thanks for your question, we will answer your question in three parts.
>
> **1.** The variational lower bound $I_{v}(X^{i}, Y^{i}; Z^{i})$ is equivalent to $E_{p(z^{i},y^{i},x^{i})}[p_{\theta}(y^{i}|x^{i},z^{i})]$+$E_{p(z^{i},x^{i})}[p_{\theta}(x^{i}|z^{i})]$, the first term $E_{p(z^{i},y^{i},x^{i})}[p_{\theta}(y^{i}|x^{i},z^{i})]$ is the negative log-likelihood of the prediction of $Y^{i}$ given $Z^{i}$ and $X^{i}$, and the second term aims to reconstruction of $X^{i}$ given $Z^{i}$. Therefore, maximizing the variational lower bound can help the model to extract sufficient information of $X^{i}$ while ensuring prediction performance.
>
> **2.** To improve the model's ability to capture cross-variable dependencies, it is crucial to simultaneously maximize the variational lower bound $I_{v}(X^{i}, Y^{i}; Z^{i})$ and minimize the upper bound $I_{vCLUB-S}(X^{o};Z^{i})$. Minimizing the variational upper bound reduces the mutual information between $Z^{i}$ and $X^{o}$, while maximizing the variational lower bound requires $Z^{i}$ to contain sufficient information for reconstructing $X^{i}$ and predicting $Y^{i}$. By striking a balance between these two objectives, CDAM can extract the most relevant information from $X^{o}$ while eliminating irrelevant information, thereby enhancing its ability to capture cross-variable dependencies.
>
> **3.** However, optimizing CDAM presents a challenge in selecting the appropriate value for the trade-off parameter $\beta$. Although experiments have shown that a larger $\beta$ can improve model performance, it also leads to a reduction in the extraction of cross-variable features. Therefore, determining the optimal value for $\beta$ requires further exploration to strike the right balance between prediction performance and cross-variable feature extraction.
>
> **Q6: Additionally on the upper bound, the authors state that they choose to adopt the sampled vCLUB and minimize $I_{vCLUB-S}(X^{o}; Z^{i})$. Could they expand on why this sampling strategy was chosen and what the actual derived upper bound might be?**
>
>
> **A6:** Thank you for your question. In comparison to the [VUB](https://arxiv.org/pdf/1612.00410.pdf) and [L1OUT](https://proceedings.mlr.press/v97/poole19a/poole19a.pdf) methods, vCLUB demonstrates a superior ability to approximate the upper bound. Consequently, we have opted to utilize vCLUB as the variational upper bound in our paper. Additionally, we have found that vCLUBSample offers greater efficiency than vCLUB. Hence, we have chosen to employ vCLUBSample in our study.

---

> ### Author Response · Authors · 2023-11-22
>
> **Q7:More comparison with statistical forecasting methods or non-transformer methods, either empirically or theoretically, could help introduce the benefits of InfoTime in a more thorough manner. While Zeng et. al (2022) and other methods are briefly discussed for their direct forecasting strategy; it could be helpful to compare InfoTime on the DLinear model to see how CAM varies from the DMS approach. Perhaps this is why RMLP was introduced, but it’s not clear what motivated the construction of RMLP and if differs at all from previous approaches which could be more suitable baseline models. While the benchmarks used are standard for transformer-based evaluation, there are other baselines that could shed more light on the characteristics of the model and whether it may help resolve some of the difficulties with transformer approaches to forecasting. Could the authors shed any light on if they considered benchmarks like M3, M4 and what might be missing from the current evaluations?**
>
>
>
> **A7:** Thank you for your valuable feedback and suggestions.
> The DMS method introduced by Zeng et al. is consistant with the 'single-step forecaster' in our paper which means that predict multi-steps in a single step. AS disscussed in our work, the single-step forecaster or DMS assumes that the future time steps are independent of each other, therefore, they have the following formulation: $p(y^{i}|z^{i},x^{i})=\prod_{j=1}^{P}p(y^{i}_{j}|z^{i},x^{i})$. However, it is important to note that future time steps are not truly independent, and although the single-step forecaster can effectively alleviate the error accumulation problem, it cannot model the correlation between future sequences, which is why we propose the TAM module to address this limitation.
>
> In our paper, we propose the linear model RMLP, which outperforms DLinear. As a result, we chose to use RMLP as the base linear model for comparison. Additionally, we also evaluated the effectiveness of InfoTime on the DLinear (linear model), TSMixer (linear model), and TimesNet (linear model) and included the Solar-energy dataset in our analysis. The results, as shown in the table below, demonstrate that InfoTime consistently improves the performance of DLinear, TSMixer, and TimesNet across different datasets. This highlights the efficacy of InfoTime in enhancing the forecasting capabilities of the Linear models and CNN-based models. Moreover, we appreciate your suggestion to include more benchmarks such as M3 and M4. We found that the M3 and M4 datasets are very different from the datasets used in our work. It is very meaningful to verify the effect of our model on these datasets in our future revision.

---

> > ### Author Response · Authors · 2023-11-22
> >
> > **A7:**
> >
> > | Dataset|Length |TimesNet | |+TAM | | +InfoTime| | TSMixer| |+TAM | | +InfoTime||DLinear ||+TAM || +InfoTime||
> > | :----:|:----:|:----:|:----:|:----:|:----:|:----:|:----:|:----:|:----:|:----:|:----:|:----:|:----:|:----:|:----:|:----:|:----:|:----:|:----:|
> > |||MSE|MAE|MSE|MAE|MSE|MAE|MSE|MAE|MSE|MAE|MSE|MAE|MSE| MAE |MSE|MAE|MSE|MAE|
> > |ETTh1|96|0.384|0.402|**0.377**|0.399|0.382|**0.391**|0.393|0.449|**0.381**|0.397|**0.381**|**0.391**|0.378|0.400| 0.373| 0.397| **0.370**|**0.394**|
> > |           |192|0.436|0.429|0.436|0.437|**0.433**|**0.421**|0.447|0.437|0.436|0.428|**0.429**|**0.419**|0.414|0.423| 0.409| 0.418| **0.408**|**0.418**|
> > |           |336|0.491|0.469|0.477|0.454|**0.467**|**0.442**|0.498|0.465|0.491|0.456|**0.467**|**0.439**|0.444|0.446| 0.440| 0.443| **0.438**|**0.440**|
> > |           |720|0.521|0.500|0.494|0.481|**0.472**|**0.463**|0.520|0.494|0.518|0.495|**0.468**|**0.462**|0.487|0.502| 0.487| 0.503| **0.482**|**0.498**|
> > |ETTh2|96|0.340|0.374|0.332|0.366|**0.288**|**0.337**|0.304|0.351|0.298|0.347|**0.292**|**0.341**|0.289|0.353| **0.285**| **0.350**| 0.287|0.353|
> > |           |192|0.402|0.414|0.397|0.408|**0.375**|**0.391**|0.396|0.405|0.388|0.402|**0.368**|**0.391**|0.383|0.418| 0.358| 0.403| **0.357**|**0.403**|
> > |           |336|0.452|0.452|0.435|0.439|**0.420**|**0.431**|0.432|0.434|0.430|0.433|**0.415**|**0.427**|0.448|0.465| 0.411| 0.443| **0.404**|**0.441**|
> > |           |720|0.462|0.468|0.437|0.449|**0.425**|**0.442**|0.437|0.450|0.425|0.444|**0.423**|**0.442**|0.605|0.551| 0.573| 0.547| **0.520**|**0.520**|
> > |Weather|96|0.172|0.220|0.163|0.213|**0.160**|**0.208**|0.163|0.211|0.161|0.211|**0.158**|**0.205**|0.176|0.237| 0.158| 0.220| **0.149**|**0.209**|
> > |           |192|0.219|0.261|0.219|0.258|**0.212**|**0.252**|0.214|0.259|0.211|0.253|**0.210**|**0.251**|0.220|0.281| 0.199| 0.260| **0.195**|**0.259**|
> > |           |336|0.280|0.306|0.272|0.296|**0.269**|**0.294**|0.274|0.300|0.267|0.300|**0.266**|**0.292**|0.265|0.319| 0.247| 0.302| **0.243**|**0.297**|
> > |           |720|0.365|0.359|0.351|0.349|**0.347**|**0.345**|0.354|0.349|**0.341**|0.346|**0.341**|**0.340**|0.323|0.362| 0.314| 0.355| **0.312**|**0.353**|
> > |Solar-Energy|96|0.236|0.276|0.225|0.265|**0.210**|**0.253**|0.228|0.281|0.220|0.267|**0.209**|**0.259**|0.221|0.292| 0.180| 0.235| **0.165**|**0.225**|
> > |                   |192|0.287|0.317|0.264|0.290|**0.241**|**0.271**|0.270|0.310|0.255|0.285|**0.246**|**0.281**|0.250|0.311| 0.194| **0.244**| **0.189**|0.254|
> > |                   |336|0.316|0.329|0.296|0.307|**0.274**|**0.291**|0.285|0.319|0.277|0.305|**0.266**|**0.298**|0.269|0.325| 0.201| **0.248**| **0.199**|0.254|
> > |                   |720|0.314|0.315|0.305|0.315|**0.280**|**0.298**|0.300|0.330|0.267|**0.292**|**0.265**|0.295|0.271|0.327| 0.205| 0.250| **0.196**|**0.248**|

---

> > > ### Author Response · Authors · 2023-11-22
> > >
> > > **Q8: Nits: sp. ‘varibales’ (pg. 5)**
> > >
> > > **A8:** Thank you for pointing out the typo, and we have modified it. We apologize for the oversight and appreciate you bringing it to our attention.

---

### Meta-Review · Area_Chair_KCii · 2023-12-06

**Metareview:**

This paper proposes cross-variable decorrelation aware feature modeling to improve the channel-mixing approaches for multivariate time series forecasting. Despite its merits, the paper has weaknesses that make it not ready for publication. As theoretical justification for the proposed scheme is lacking, comprehensive experimentation with detailed analysis is expected to make the claims more convincing. Perhaps that is why the reviewers are not willing to cast a stronger vote for acceptance. The authors are encouraged to take the comments and suggestions of the reviewers seriously to improve their paper to appeal better to the research community.

**Justification For Why Not Higher Score:**

Stronger experiments to substantiate the claims are needed. Currently no reviewer is willing to champion the paper for acceptance beyond a lukewarm support.

**Justification For Why Not Lower Score:**

N/A

---

### Decision · Program_Chairs · 2024-01-16

Reject